# Distinct 'safe zones' at the nuclear envelope ensure robust replication of heterochromatic chromosome regions

**Hani Ebrahimi[†], Hirohisa Masuda[†], Devanshi Jain[‡], Julia Promisel Cooper***

Telomere Biology Section, Laboratory of Biochemistry and Molecular Biology, Center for Cancer Research, National Cancer Institute, Bethesda, United States

**Abstract** Chromosome replication and transcription occur within a complex nuclear milieu whose functional subdomains are beginning to be mapped out. Here we delineate distinct domains of the fission yeast nuclear envelope (NE), focusing on regions enriched for the inner NE protein, Bqt4, or the lamin interacting domain protein, Lem2. Bqt4 is relatively mobile around the NE and acts in two capacities. First, Bqt4 tethers chromosome termini and the *mat* locus to the NE specifically while these regions are replicating. This positioning is required for accurate heterochromatin replication. Second, Bqt4 mobilizes a subset of Lem2 molecules around the NE to promote pericentric heterochromatin maintenance. Opposing Bqt4-dependent Lem2 mobility are factors that stabilize Lem2 beneath the centrosome, where Lem2 plays a crucial role in kinetochore maintenance. Our data prompt a model in which Bqt4-rich nuclear subdomains are 'safe zones' in which collisions between transcription and replication are averted and heterochromatin is reassembled faithfully.

DOI: https://doi.org/10.7554/eLife.32911.001

**\*For correspondence:**
julie.cooper@nih.gov

[†]These authors contributed equally to this work

**Present address:** [‡]Memorial Sloan Kettering Cancer Center, New York, United States

**Competing interests:** The authors declare that no competing interests exist.

## Introduction

Within the nucleus, specific chromosome regions tend to localize to the periphery where they contact components of the nuclear envelope (NE) (*Bickmore, 2013*). Peripherally positioned chromatin tends to be enriched with repressive histone modifications associated with transcriptional silencing, while actively transcribing gene-rich regions tend to reside at internal positions within the nucleus. However, nuclear pore complexes (NPCs) at the periphery transiently host transcriptionally active regions (*Capelson et al., 2010*; *Ishii et al., 2002*; *Lemaître and Bickmore, 2015*). Therefore, the NE appears to harbor at least two functionally distinct domain types.

Heterochromatic regions generally contain hypoacetylated nucleosomes with relatively low turnover rates, and histone H3 that is di- or tri-methylated at Lysine 9 (H3K9me) (*Aygün et al., 2013*; *Rea et al., 2000*). Chromatin regions enriched with H3K9me recruit protein complexes that suppress gene transcription; crucial to this network are histone deacetylases (HDACs). Studies in the fission yeast *S. pombe* have been seminal to our understanding of conserved mechanisms underlying heterochromatin formation and function (*Allshire and Ekwall, 2015*; *Grewal and Jia, 2007*; *Martienssen and Moazed, 2015*; *Motamedi et al., 2008*; *Woolcock et al., 2011*). Clr4 is the sole histone *S. pombe* H3K9 methyltransferase and is critical for establishment of heterochromatin (*Nakayama et al., 2001*; *Ragunathan et al., 2015*). H3K9me-containing nucleosomes provide a binding platform for HP1 family chromodomain proteins, including Swi6. HP1-mediated recruitment of SHREC, which harbors both a HDAC and a Snf2-like nucleosome remodeler (*Sugiyama et al., 2007*), creates a positive feedback loop to ensure recursive H3K9 methylation, recruitment of HP1 and spreading of SHREC along the silenced DNA region (*Motamedi et al., 2008*; *Zhang et al., 2008*). The conserved JmjC protein Epe1 prevents repressive marks from spreading beyond the

limits of the heterochromatic domain (*Shimada et al., 2009*; *Trewick et al., 2007*; *Zofall and Grewal, 2006*).

The nuclear RNAi pathway triggers heterochromatin assembly at several loci. Transcripts of such loci are converted to dsRNA by the RNA dependent RNA polymerase or convergent transcription, creating substrates for the Dicer (Dcr1) ribonuclease, which cleaves dsRNA to produce siRNAs. Once loaded into the Argonaute complex (RITS), these siRNAs target RITS to the chromosome *via* homology to nascent transcripts, eliciting H3K9 methylation, HP1 binding and transcriptional silencing. Dicer also acts to evict RNA polymerase II (RNAP2) from highly transcribed regions and from repetitive DNA regions like the tDNA, rDNA and pericentric repeats. Dicer-mediated release of RNAP2 promotes genome stability by limiting collisions between the transcription and replication machineries (*Castel et al., 2014*; *Kloc et al., 2008*; *Zaratiegui et al., 2011*).

Centromeres are among the most prominent heterochromatic regions. The centromeric central core, upon which the kinetochore is assembled, is flanked by heterochromatic repeats comprising the innermost repeats (*imr*) and the outer repeats (*otr*) (*Takahashi et al., 1992*; *Steiner et al., 1993*; *Wood et al., 2002*). This organization, in which a core platform for kinetochore assembly is flanked by heterochromatic repeats, is conserved in vertebrates (*Kniola et al., 2001*). HP1-containing pericentric heterochromatin recruits cohesin to the outer repeats (*Nonaka et al., 2002*), thereby ensuring faithful chromosome segregation (*Bernard et al., 2001*). Pericentric heterochromatin also plays a crucial role in the de novo assembly of functional CenpA-containing centromeres capable of kinetochore assembly (*Folco et al., 2008*). Once a functional centromere is assembled, however, heterochromatin becomes dispensable for kinetochore maintenance.

Heterochromatin also extends inwards from chromosomal termini, for ~30 kb in *S. pombe*. Reporter genes inserted at these terminal regions are transcriptionally silenced (*Cooper et al., 1997*; *Kanoh et al., 2005*). Telomeric DNA, consisting of double-stranded (ds) G-rich repeats with a terminal single-stranded (ss) G-rich 3' overhang, provides the platform for a sextet of proteins known collectively as shelterin (*de Lange, 2009*), which regulates telomerase activity and prevents chromosome ends from being treated as DNA double strand breaks. Heterochromatin is not required for these crucial telomere functions (*Khair et al., 2010*; *Tuzon et al., 2004*). However, in survivors of telomerase loss, heterochromatin is critical for either protecting chromosome ends *via* 'HAATI' (*Jain et al., 2010*) or conversely, for inhibiting the 'ALT' pathway of continual break induced replication characterized in telomerase-negative cancers (*Benetti et al., 2008*; *Benetti et al., 2007*). The fission yeast shelterin component Taz1 (ortholog of human TRF1 and TRF2) binds directly to telomeric dsDNA; Rap1 binds Taz1 (and, at least in the absence of Taz1, weakly binds other shelterin proteins) and directly interacts with the inner NE protein Bqt4 (*Chikashige et al., 2009*), promoting telomere localization to the NE. Telomere tethering is also promoted in a Bqt4 independent manner by the conserved Fun30 chromatin remodeler Fft3 (*Steglich et al., 2015*). Nonetheless, telomeres are not always tethered; for instance, dislodgment of telomeres from the NE prior to mitosis appears to contribute to accurate chromosome segregation (*Fujita et al., 2012*).

In addition to Bqt4 and the NPCs, other key fission yeast NE complexes include the 'linker of nucleoskeleton and cytoskeleton complex' (LINC) and the Lap2/Emerin/Man1 (LEM) family of lamin-associated proteins (*Gonzalez et al., 2012*). LINC comprises the conserved SUN- (Sad1) and KASH-domain (Kms1/Kms2) proteins that form a trans-NE link from nucleoplasm to cytoplasm. Kms2 interacts with the centrosome (the spindle pole body or SPB), which resides on the outer surface of the NE. The nucleoplasmic extension of LINC (Sad1) interacts with centromeres, which are clustered at the NE beneath the SPB (*Fernández-Álvarez et al., 2016*; *Hagan and Yanagida, 1995*; *Scherthan et al., 2011*). Two LEM proteins, Lem2 and Man1, fulfill lamin-like functions despite the absence of canonical nuclear lamina in fission yeast (*Cai et al., 2001*; *Gonzalez et al., 2012*; *Lee et al., 2001*). Man1 has been shown to interact with regions 50–100 Kb from telomeres (*Steglich et al., 2012*), while Lem2 appears to be sequestered near centromeres (*Hiraoka et al., 2011*). Recent studies have identified Lem2 as a regulator of heterochromatin silencing at pericentric regions (*Barrales et al., 2016*; *Tange et al., 2016*).

Notwithstanding the tendencies of specific chromatin regions to associate with the NE, the functions of this association in most instances remain to be elucidated. We identified Bqt4 in a screen (to be described elsewhere) for genes involved in maintaining the HAATI mode of telomerase-negative survival (*Jain et al., 2010*), in which heterochromatic rDNA repeats translocate to all chromosome ends and acquire end protection capacity. In exploring the function of Bqt4 in wild type (*wt*) cells,

we uncovered two classes of roles for Bqt4, one in directly regulating a subset of chromatin regions specifically during S phase, and another in mobilizing Lem2, which in turn has distinct roles in maintenance of the pericentric heterochromatin and the centromeric central core. Along with the S-phase specificity of Bqt4's role in chromatin localization, we report several *bqt4Δ* phenotypes and genetic interactions that implicate Bqt4-rich microdomains as specialized 'safe zones' that prevent collisions between transcription and replication machineries, and thereby confer accurate maintenance of the heterochromatic state through successive generations.

## Results

### NE proteins organize into distinct domains

To chart the organization of key proteins at the NE, we used super-resolution fluorescence imaging (the OMX system) (*Dobbie et al., 2011*) to view endogenously tagged Bqt4, Man1 and Lem2 (see Materials and methods). Consistent with previously described electron microscopy images (*Chikashige et al., 2009*), we observe that Bqt4 does not uniformly distribute around the NE but rather appears in puncta, indicating spatial heterogeneity in NE composition (*Figure 1A*). To address the composition of regions that lack detectable Bqt4 (dark areas in *Figure 1A*), we simultaneously imaged Bqt4 and Man1 (*Figure 1B*) or Bqt4 and Lem2 (*Figure 1C*). Bqt4 and Man1 signals occasionally overlap but more often, they do not (*Figure 1C*, *Figure 1—figure supplement 1A*). Domain partitioning is also seen when simultaneously viewing Bqt4 and Lem2 (*Figure 1C*, *Figure 1—figure supplement 1A*).

The most intense Lem2 pool appears just beneath the SPB, while a less prominent subset of Lem2 molecules localizes to a ring around NE (*Figure 1E*). Remarkably, the ring of Lem2 is diminished to near-invisibility in cells lacking Bqt4, while a single Lem2 focus remains (*Figure 1E* and *Figure 1—figure supplement 2A*), colocalizing with the SPB component Pcp1 (*Figure 1—figure supplement 2A*) (*Gonzalez et al., 2012*). In contrast, the localization patterns of the NPC protein Nup107 (*Baï et al., 2004*), the NE component Cut11 (*West et al., 1998*), and Man1 are all independent of Bqt4 (*Figure 1—figure supplement 2B*). Hence, Bqt4 is specifically required for distribution of Lem2 around the NE.

### Bqt4 is highly mobile within the NE

The distinct localization patterns for Bqt4, Man1 and Lem2, along with the observation that Bqt4 controls the mobility of Lem2 around the NE, suggest that protein dynamics differ among these NE components and in different NE regions. Indeed, variable levels of dynamicity have previously been reported for NE proteins (*Ellenberg et al., 1997*; *Rabut et al., 2004*). To explore this, we used fluorescence recovery after photobleaching (FRAP) to assess turnover rates. *Figure 2A* shows a representative nucleus harboring Bqt4-GFP. After capturing a reference 'pre-bleach' image, we photobleached a region of interest (ROI) encompassing a segment of the NE (*Figure 2A*). Bqt4-GFP signal inside the photobleached window recovers within ~12 s (*Figure 2B*; half-life 5.029 s). Moreover, the reappearance of Bqt4-GFP signal within the ROI consistently shows a distinct recovery pattern; rather than uniformly recovering throughout the bleached NE segment, the signal appears progressively from the flanking NE edges to the center of the bleached region (*Figure 2A*), suggesting that dissociated Bqt4-GFP is replenished by movement around the NE. While Bqt4 in the unbleached portion of the NE may be continually replenished from nucleoplasmic pools, the described recovery pattern cannot be explained by exchange with a non-membrane bound pool of Bqt4.

The substantial reduction in Lem2-GFP levels encircling the NE in *bqt4Δ* cells prevents the robust use of FRAP for measuring Lem2 turnover around the NE in this setting. However, we were able to measure the turnover rate of Lem2-GFP localized beneath the SPB in both *wt* and *bqt4Δ* cells. At this site, the half life of Lem2 residency remains unaffected upon Bqt4 loss (*Figure 2C*; Half life, *wt*: 23.11 s, *bqt4Δ*: 23.99 s). Hence, the two distinct pools of Lem2, beneath the SPB and around the NE, are controlled by different mechanisms, with Bqt4 controlling the pool of Lem2 that is distributed away from the SPB region around the NE.

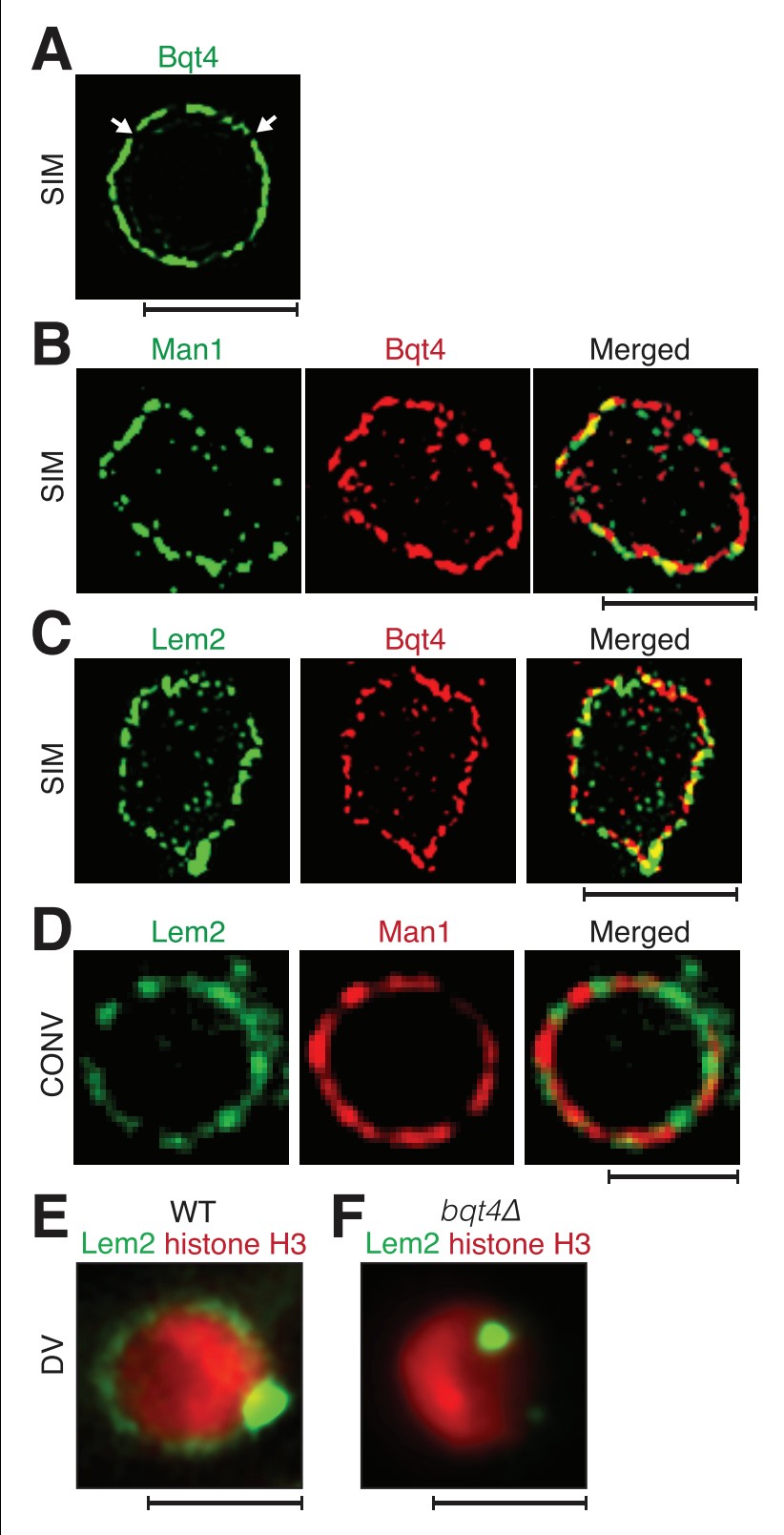

**Figure 1.** Inner NE proteins are sequestered into distinct microdomains. (**A**) Representative super-resolution microscopy (structured illumination, SIM) image of a live cell harboring Bqt4-GFP. Lateral resolution is 120–150 nm. Arrows indicate regions with relatively low abundance of detectable fluorophore. Scale bars represent 2 μm in all images. (**B**) SIM images of representative fixed cells expressing Man1-GFP and Bqt4-mCherry, and (**C**) Lem2-GFP

*Figure 1 continued on next page*

*Figure 1 continued*

and Bqt4-mCherry. Colocalization quantitations and procedures are described in *Figure 1—figure supplement 1A*. (D) DeltaVision deconvolution fluorescence imaging of live cells harboring Lem2-GFP and Man1-tdTomato. Quantitation is shown in *Figure 1—figure supplement 1C*. (E) A representative single z-plane image of a live *wt* cell expressing Lem2-GFP and histone H3-mCherry. Lem2-GFP is detectable around the NE with an intense dot beneath the SPB. (F) In the absence of Bqt4, Lem2 fails to encircle the NE but remains beneath the SPB. Lateral resolution for (E–F) ~300 nm.

DOI: https://doi.org/10.7554/eLife.32911.002

The following figure supplements are available for figure 1:

**Figure supplement 1.** Quantitation of NE protein colocalizations.

DOI: https://doi.org/10.7554/eLife.32911.003

**Figure supplement 2.** Bqt4 controls localization of Lem2 but not Man1, Nup107 or Cut11.

DOI: https://doi.org/10.7554/eLife.32911.004

## Lem2 localization beneath the SPB is regulated by the LINC-interacting protein Csi1

As Lem2 concentrates beneath the SPB, we conjectured that the LINC complex controls Lem2 accumulation at this site. Csi1 (chromosome segregation impaired protein 1) associates with Sad1 and promotes the robust association of centromeres with the LINC complex (*Hou et al., 2012*), making Csi1 a candidate regulator of Lem2 localization to this region. While the intense Lem2 focus colocalizes with the SPB component Sid4 in *wt* cells (*Chang and Gould, 2000*), this colocalization is abolished in 85% of *csi1Δ* cells as evinced by the absence of a Lem2-GFP dot beneath the SPB (*Figure 3A*). Notably, however, the ring of Lem2-GFP around the NE remains in *csi1Δ* cells. Therefore, Csi1 either recruits or stabilizes Lem2 beneath the SPB.

As Bqt4 promotes the distribution of Lem2 around NE while Csi1 concentrates Lem2 beneath the SPB, we wondered where Lem2 would localize when both these factors are missing. In cells lacking both Bqt4 and Csi1, Lem2-GFP is greatly diminished from the ring around the NE but still strongly detectable beneath the SPB (*Figure 3B*). However, FRAP analysis reveals that the Lem2 turnover rate is substantially increased in *csi1Δ bqt4Δ* cells relative to *wt* or *bqt4Δ* cells (*Figure 3C*). Hence, while the initial Lem2 recruitment to the LINC region beneath the SPB is controlled independently of Bqt4 and Csi1, Csi1 is required to stabilize Lem2 at this region; this stabilization is required to counterbalance the Bqt4-dependent mobilization of Lem2 away from the LINC region.

## Bqt4 tethers telomeres and the *mat* locus specifically while they are replicating

Bqt4 has been implicated in the interphase positioning of telomeres, which form two to three clusters at the NE (*Funabiki et al., 1993*; *Chikashige et al., 2009*). To assess whether telomeres localize specifically to Bqt4-containing regions, we visualized telomeres in cells harboring the various tagged NE proteins (*Figure 3D–F*). Consistent with previous observations (*Dehé et al., 2012*; *Funabiki et al., 1993*), we observe two to three Taz1 foci in each nucleus. *Figure 3D* shows a single focal plane that contains a single Taz1 focus; this telomeric focus colocalizes with Bqt4-GFP (*Figure 3D*). In contrast, telomeres show markedly lower degrees of colocalization with Man1 or Lem2 (*Figure 3E–G*, *Figure 3—figure supplement 1*). Therefore, telomeres have a higher propensity to localize to Bqt4-rich than Man1- or Lem2-rich NE domains.

Centromeres localize to the NE during interphase, forming a cluster beneath the SPB (*Funabiki et al., 1993*). The intense focus of Lem2 detectable in this region (*Figure 4A*) is consistent with ChIP experiments showing an interaction between Lem2 and centromeres (*Barrales et al., 2016*; *Tange et al., 2016*). To investigate the exclusivity of centromere-Lem2 colocalization, we visualized endogenously mCherry-tagged Mis6, an inner kinetochore component (*Takahashi et al., 2000*). Consistent with previous observations (*Hiraoka et al., 2011*), Mis6 sequesters within a Lem2-enriched domain beneath the SPB (*Figure 4A*, *Figure 4—figure supplement 1*). In contrast, Mis6 does not colocalize with Man1 (*Figure 4B*, quantified in *Figure 4—figure supplement 1B*). To observe the centromeric DNA sequences immediately surrounding the kinetochore region, we imaged cells that contain mCherry-tagged tet repressor (tetR-mCherry) and an array of tetracycline operators (tetO) inserted within the *imr* of centromere I. The marked *imr* and the SPB (Sid4-GFP)

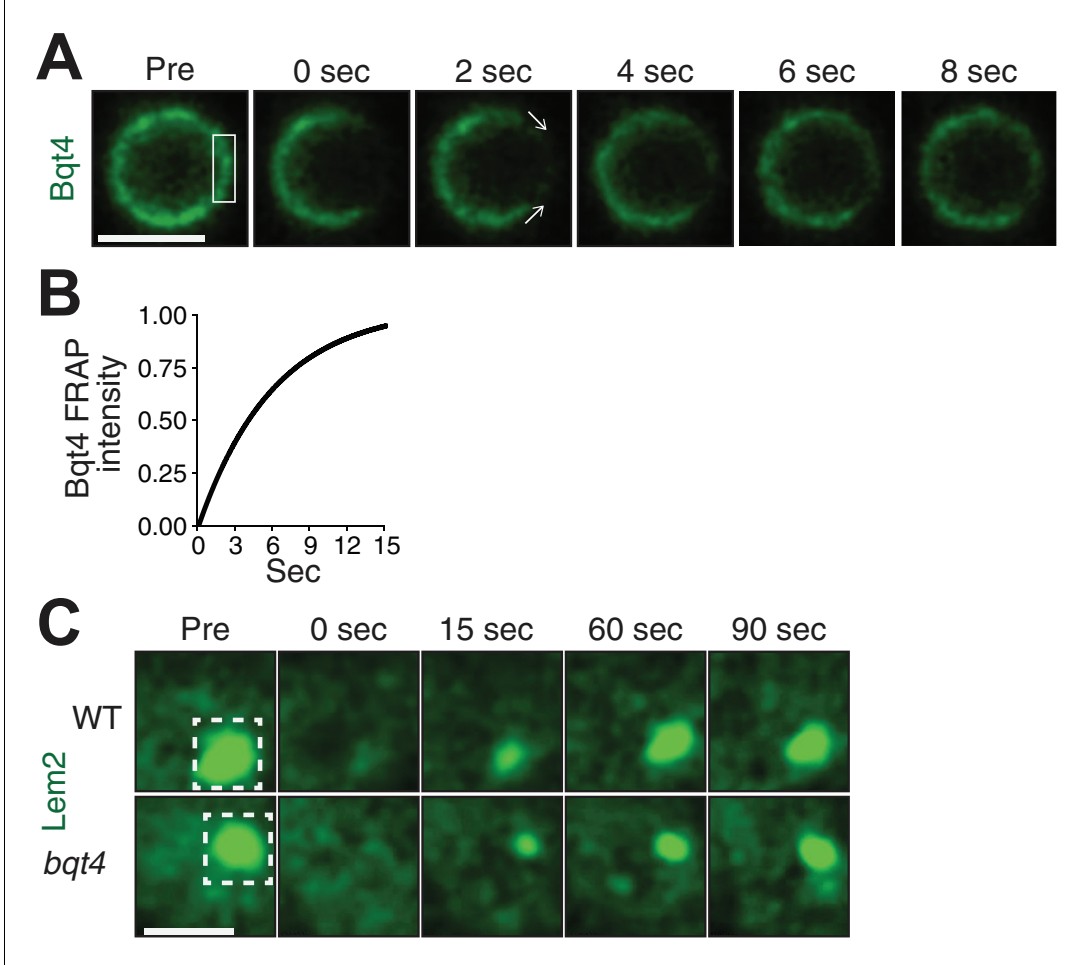

**Figure 2.** Bqt4 mobilizes Lem2 around the NE. (**A**) Representative time-lapse series of a FRAP experiment. 'Pre' shows a control image captured immediately prior to photo-bleaching. The white box indicates the ROI targeted for photobleaching. Time 0 is taken immediately after photo-bleaching. Images were captured every 2 s. Quantitation was performed by measuring sum of pixel values inside ROI, deducting background signal (a region next to the nucleus with the same size as the ROI), and dividing by ROI signal in 'Pre' image. Scale bars here and in (**C**) represent two microns. (**B**) The normalized fluorescence intensities (described in A) from the imaged cells were averaged for each corresponding timepoint, and a curve fit was plotted using Prism (GraphPad Software). The y-axis indicates normalized intensities relative to 'Pre' intensity set to 1.0 (half-life 5.03 s, n = 13). (**C**) Cells expressing Lem2-GFP were subjected to FRAP as in A, except that the time-lapse interval was increased to 15 s. The ROI (dashed box in 'Pre') was drawn around a region covering the visible Lem2-GFP dot. The Lem2-GFP ring around the nucleus, visible by other imaging systems (**Figure 1E,F**), was not detectable with the specific FRAP system and settings used here. Quantitations are shown in **Figure 3C**.
DOI: https://doi.org/10.7554/eLife.32911.005

colonize in 100% of cells (**Figure 4—figure supplement 1C**). We also imaged cells that contain LacI-CFP and an array of Lac operators (LacO) integrated ~10 Kb away from the *otr* of centromere III (**Ding et al., 2004**). In contrast to the central *imr*, this distal *otr* region colocalizes with the SPB in only 12% of cells (**Figure 4—figure supplement 1C**); in the majority of cells, *otr* appears adjacent to, rather than superimposed upon, the SPB. Hence, the central core region and *imr* localize exclusively to the SPB region, while the pericentric *otr* sequences associate with NE regions away from the SPB. Collectively, these observations reveal a NE with distinct microdomains that are exclusively enriched with Man1, Lem2 or Bqt4, and interact with distinct chromosomal regions.

The specificity of Bqt4-mediated NE localization for telomeres might be envisioned to stem from their heterochromatic nature (**Cooper et al., 1997**). However, H3K9me-containing heterochromatin is dispensable for telomeric tethering to the NE, as Tel1L is not dislodged from the NE in *clr4Δ* cells (**Figure 4—figure supplement 1D**). Hence, although *clr4+* deletion was reported to delocalize the *mat* locus (**Alfredsson-Timmins et al., 2007**), telomeres are tethered to the NE in a H3K9me-

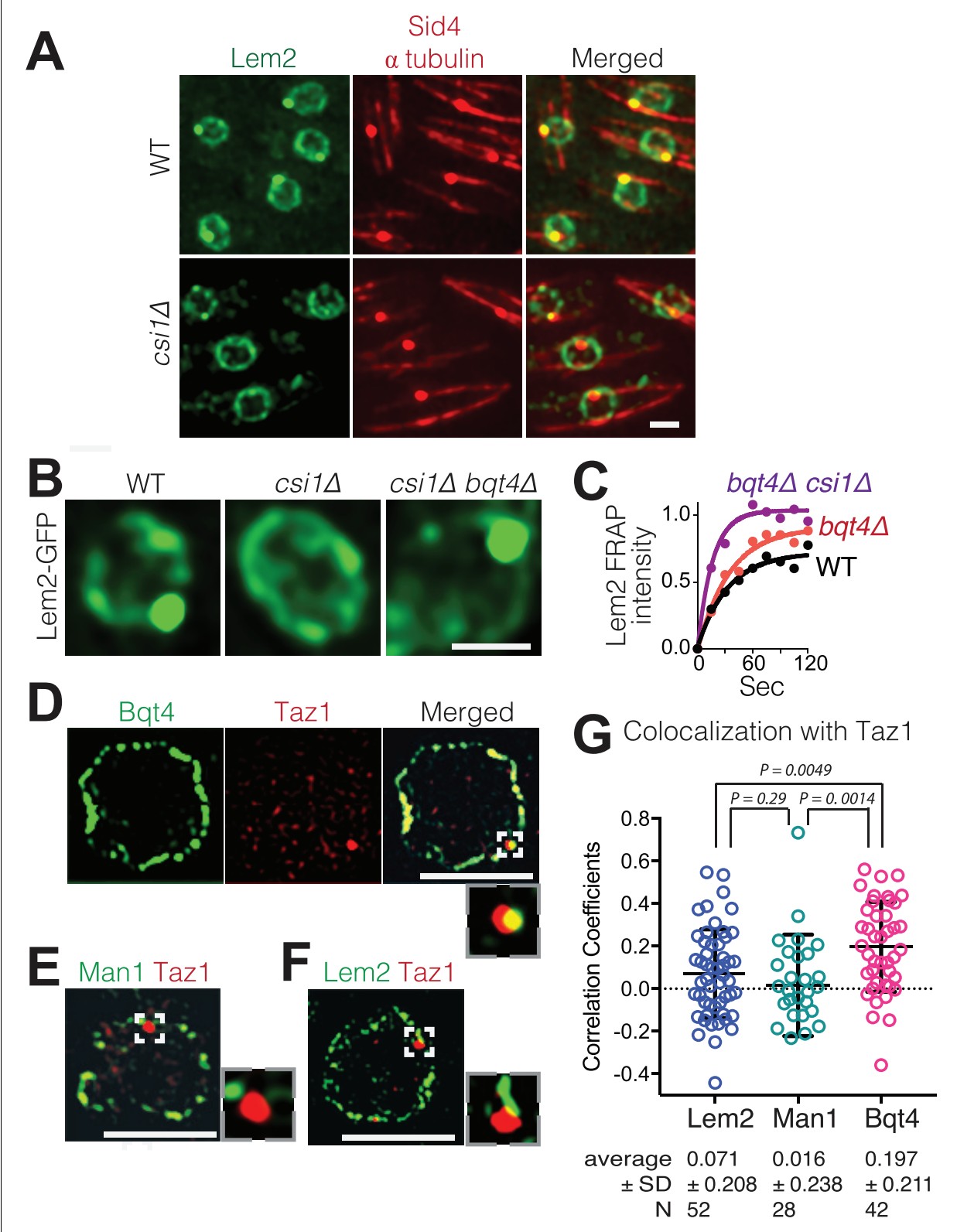

**Figure 3.** Csi1 stabilizes Lem2 beneath the SPB while Bqt4 moves Lem2 around the NE. (**A**) High-resolution images (~300 nm lateral resolution) show that Lem2 localization beneath the SPB requires Csi1. Lem2-GFP and Sid4-mRFP overlap in all WT cells, but fail to overlap in 85% of *csi1Δ* cells. Scale bars represent two microns in all images. (**B**) Deletion of *bqt4+* restores detectable Lem2-GFP dot at the SPB in *csi1Δ* cells. (**C**) FRAP of the visible Lem2-GFP dot as described in *Figure 2C*. Non-linear fit of the fluorescence recoveries for the indicated strains show curtailed half-life for Lem2-GFP at
*Figure 3 continued on next page*

*Figure 3 continued*

the SPB in the *bqt4Δ csi1Δ* double deletion strains. Half-lives: WT 23.11 s, n = 12; *bqt4Δ* half-life 23.99 s, n = 10; *bqt4Δ csi1Δ* half-life 11.24 s, n = 18. Mobile fraction (plateau) for WT: 72%, *bqt4Δ*: 91%, *bqt4Δcsi1Δ*: 100%. Goodness of fit ($R^2$) for WT: 0.961, *bqt4Δ*: 0.974, *bqt4Δcsi1Δ*: 0.956. FRAP could not be performed in the *csi1Δ* background due to lack of visible Lem2 at the SPB. (D) SIM (120–150 nm lateral resolution) images of representative fixed cells expressing Bqt4-GFP and Taz1-mCherry, (E) Man1-GFP and Taz1-mCherry, and (F) Lem2-GFP and Taz1-mCherry. Insets show magnifications of boxed regions. (G) Co-localization between Taz1 and the indicated proteins was quantified as Pearson correlation coefficients, using the Applied Biosystems softWorX software.

DOI: https://doi.org/10.7554/eLife.32911.006

The following figure supplement is available for figure 3:

**Figure supplement 1.** Additional examples of simultaneous imaging of Lem2-GFP and Taz1-mCherry by SIM microscopy.

DOI: https://doi.org/10.7554/eLife.32911.007

independent manner. This is consistent with the observation that *taz1Δ* telomeres, in which silencing and H3K9 methylation are vastly reduced, nonetheless retain NE localization (*Zaaijer et al., 2016*).

The distinct localization patterns of telomeres and centromeres with respect to Bqt4 and Lem2 prompted us to explore the functional significance of Bqt4- and Lem2-microdomains. First, we examined telomere positioning as a function of cell cycle stage using cell shape as indicator. Soon after mitotic nuclear division, fission yeast cells undergo S-phase, before cytokinesis is complete; therefore, binucleate cells are in early/mid S-phase. As cells progress from late S phase to G2, cytokinesis gives rise to the two mononucleate daughter cells. Accordingly, we scored interphase binucleate cells as early/mid-S phase and mononucleate cells as late S/G2. Telomere 1-left (Tel1L) was visualized by a LacO array integrated at the *sod2+* locus (*Ding et al., 2004*) in the centromere-proximal subtelomeric region (~50 Kb from chromosome end) in cells expressing LacI-GFP (*Ding et al., 2004*), hereafter referred to as Tel1L lacO/I-GFP. The NE was viewed by tagging the NE protein Cut11 with mCherry to allow measurement of the distance between Tel1L-lacO/I-GFP and the NE. The peripheral zone was defined as the outer third of the nuclear volume; a randomly positioned sequence will localize to this zone in ~33% of cells (*Meister et al., 2010*); statistics for all measurements are shown in *Table 1*. In 80% of *wt* settings, Tel1L localizes to the nuclear periphery (*Figure 4C*) whether early/mid S-phase or late S/G2. This localization is disrupted in a cell cycle dependent manner in *bqt4Δ* cells. While Tel1L is at the periphery in ~60% of early/mid-S phase *bqt4Δ* cells, dislodgement is substantial in late S/G2 phase, with only ~48% localization at the NE (*Figure 4C*). Hence, the role of Bqt4 in localizing telomeres to the NE is more pronounced during late S/G2 than in early S phase.

As Lem2 is lost from SPB-distal regions of the NE in a *bqt4Δ* setting, we examined telomere positioning in *lem2Δ* cells. In contrast to *bqt4Δ* cells, Tel1L remains at the periphery throughout the cell cycle in *lem2Δ* cells (*Figure 4C*). Hence, loss of tethering in a *bqt4Δ* setting is not a function of losing Lem2 from telomeric tethering sites. Centromeres remain localized beneath the SPB in the absence of either Bqt4 or Lem2 (data not shown). Therefore, any role of Lem2 in tethering chromosomal loci is redundant with other mechanisms (*Barrales et al., 2016*).

To determine whether the rules governing Tel1L localization are shared by all telomeres, we monitored endogenously tagged and functional Taz1-GFP, which binds directly to all telomeres. Taz1 localizes strongly to the periphery and is delocalized only during late S/G2 in the absence of Bqt4 (*Figure 4D*). The mating type locus (*mat*) is another prominent genomic region that positions at the NE (*Alfredsson-Timmins et al., 2007*; *Noma et al., 2006*). To determine whether this localization requires Bqt4, we utilized a strain in which the lacO/I array is inserted at *mat*. While peripheral localization was observed during both S and G2-phase in *wt* cells, *mat* is dislodged from the NE in early/mid-S phase in *bqt4Δ* cells (*Figure 4D*). This dislodgment is transient, as *mat* re-localizes to the NE in late S/G2. In contrast, a control strain harboring a lacO/I array adjacent to a euchromatic locus (*cut3*) on the left arm of chromosome II (Chr II) (*Ding et al., 2004*) shows random positioning throughout the nucleus in both *wt* and *bqt4Δ* settings (*Figure 4D*).

The *mat* locus replicates in early S phase, while telomeres replicate in late S phase (*Hayano et al., 2012*; *Hayashi et al., 2009*; *Kim et al., 2003*). This temporal discord in the replication of telomeres *versus* the *mat* locus correlates with the temporal discord in dislodgment of these loci from the nuclear periphery in a *bqt4Δ* setting. Hence, we postulated that Bqt4 is necessary for tethering these loci specifically while they undergo DNA replication. To assess this possibility, we abolished the late specificity of telomere replication by deleting the gene encoding Taz1; in a *taz1Δ*

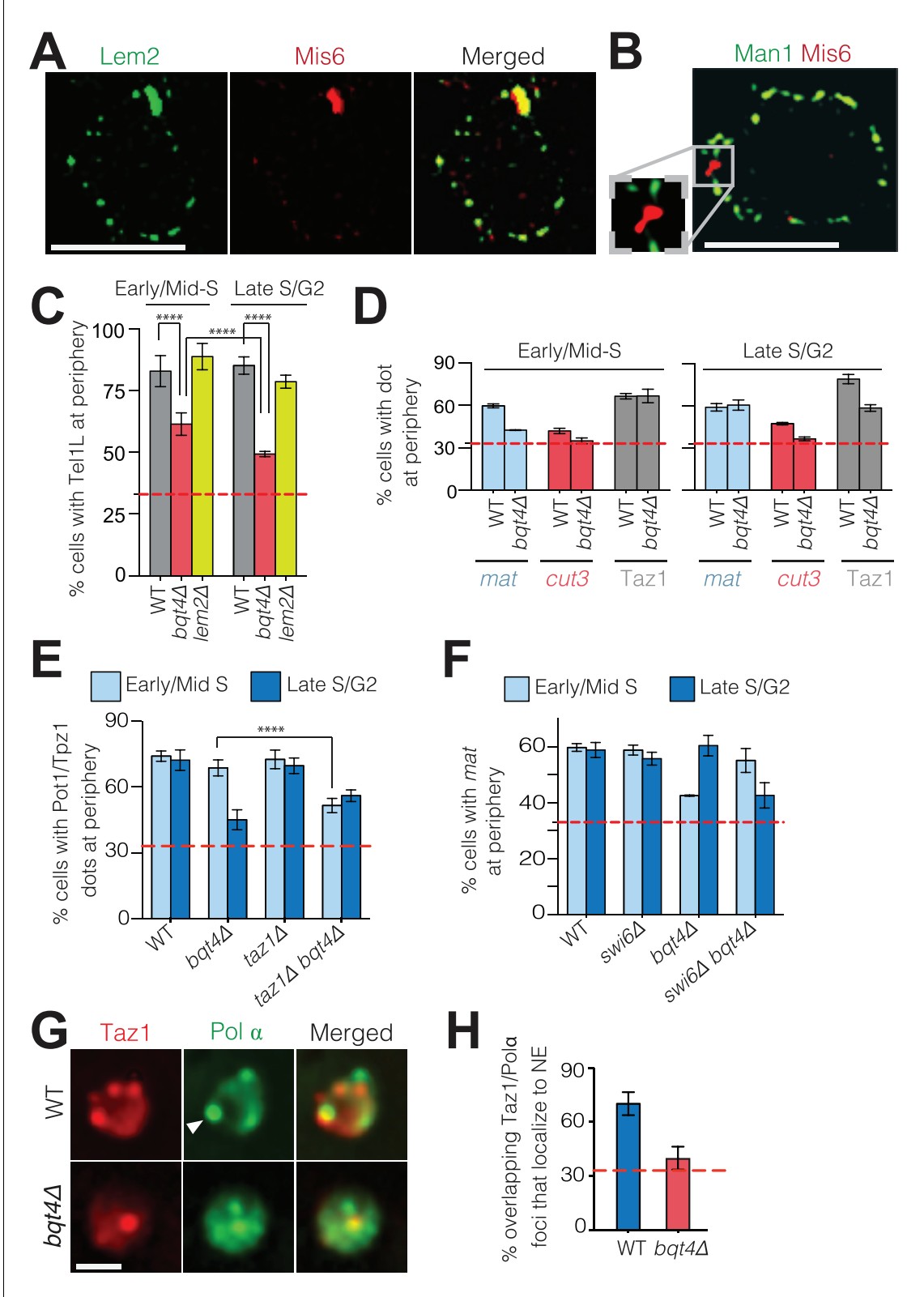

**Figure 4.** Bqt4 tethers telomeres and the *mat* locus when undergoing DNA replication. (**A**) SIM images of representative fixed cells expressing Lem2-GFP and Mis6-mCherry, and (**B**) Man1-GFP and Mis6-mCherry. Inset in (**B**) shows magnification of the boxed region. (**C**) Bqt4 regulates telomere positioning while Lem2 does not. Snapshots of live cells harboring Tel1L-lacO/I-GFP and Cut3-mCherry (to mark the NE) were captured, the distance between the telomere and NE measured, and distances categorized based on the nuclear zoning assay (see Materials and methods) in which the outer

*Figure 4 continued on next page*

*Figure 4 continued*

third of the nuclear volume is considered the periphery. Dashed red line indicates the level of peripheral zoning expected for random localization within the nucleus (33%). The Y-axis indicates percent of imaged cells in which telomere-NE distance is categorized as Zone I (the most peripheral zone with a maximum distance of ~0.22 µm from the NE). *Table 1* shows statistics for the data here and in (**D–F**). Error bars indicate SD. **\*\*\*\*** indicates p≤0.0001 as determined by a Student's t-test. (**D**) Imaging, quantitation and plotting as in C. The *mat* locus and *cut3* gene were visualized *via* lacO/I arrays inserted <40 kb away; telomere clusters were visualized *via* Taz1-mCherry. (**E–F**) Imaging and quantitation as described in A (n > 80). (**G**) Colocalization of Taz1-mCherry and Pol α-GFP in the indicated strains was assessed using automated image-analysis on 800–1000 cells, utilizing a MATLAB script that uses a constant threshold level for detecting dots and assigning colocalization, and measures distance to the edge of the nucleus. Arrowhead indicates a site of Polα/telomere colocalization. (**H**) Data from G were plotted as described in C.

DOI: https://doi.org/10.7554/eLife.32911.008

The following figure supplement is available for figure 4:

**Figure supplement 1.** Imaging and quantitation of NE protein colocalization with centromeres and pericentromeres.

DOI: https://doi.org/10.7554/eLife.32911.009

background, telomeres replicate throughout the cell cycle (*Dehé et al., 2012*; *Tazumi et al., 2012*). We visualized *taz1Δ* telomeres by GFP tagging the shelterin components Pot1 or Tpz1, which bind the ss telomeric overhang and retain telomere localization in the absence of Taz1 (*Baumann and*

**Table 1.** Statistical significance of chromosomal localization data in *Figure 4*.

$n_1$ and $n_2$ denote total number of scored loci in cells from two biological replicates (independent strains). *P* values, from two-tailed Student's t-tests, indicate the significance of the difference between the observed number of loci in Zone one and the expected random distribution.

| Locus genotype | Early/Mid-s | Late S/G |
|---|---|---|
| Tel 1L<br>WT | $n_1 = 112,, n_2 = 93$<br>p<0.000001 | $n_1 = 151, n_2 = 180$<br>p<0.000001 |
| Tel 1L<br>*bqt4Δ* | $n_1 = 110, n_2 = 103$<br>p<0.000001 | $n_1 = 143, n_2 = 112$<br>p=0.00986 |
| Tel 1L<br>*lem2 Δ* | $n_1 = 110, n_2 = 86$<br>p<0.000001 | $n_1 = 96, n_2 = 160$<br>p<0.000001 |
| *mat*<br>WT | $n_1 = 55, n_2 = 49$<br>$p=5.0 \times 10^{-5}$ | $n_1 = 85, n_2 = 95$<br>$p=1.0 \times 10^{-5}$ |
| *mat*<br>*bqt4Δ* | $n_1 = 70, n_2 = 52$<br>p=0.79551 | $n_1 = 91, n_2 = 90$<br>p<0.0001 |
| *cut3*<br>WT | $n_1 = 86, n_2 = 73$<br>p=0.0459 | $n_1 = 57, n_2 = 113$<br>p=0.97412 |
| *cut3*<br>*bqt4Δ* | $n_1 = 94, n_2 = 90$<br>p=0.99545 | $n_1 = 78, n_2 = 125$<br>p=0.99801 |
| Taz1<br>WT | $n_1 = 51, n_2 = 74$<br>p<0.000001 | $n_1 = 85, n_2 = 87$<br>p<0.000001 |
| Taz1<br>*bqt4Δ* | $n_1 = 67, n_2 = 93$<br>p<0.000001 | $n_1 = 88, n_2 = 86$<br>$p=5.0 \times 10^{-5}$ |
| Pot1<br>WT | $n_1 = 79, n_2 = 65$<br>p<0.000001 | $n_1 = 54, n_2 = 68$<br>p<0.000001 |
| Pot1<br>*bqt4Δ* | $n_1 = 64, n_2 = 96$<br>p<0.000001 | $n_1 = 59, n_2 = 81$<br>p=0.0293 |
| Pot1<br>*taz1Δ* | $n_1 = 60, n_2 = 63$<br>p<0.000001 | $n_1 = 88, n_2 = 96$<br>p<0.000001 |
| Pot1<br>*taz1Δ bqt4Δ* | $n_1 = 65, n_2 = 100$<br>p=0.00025 | $n_1 = 64, n_2 = 149$<br>$p=2.0 \times 10^{-5}$ |
| *mat*<br>*swi6Δ* | $n_1 = 72, n_2 = 60$<br>$p=1.0 \times 10^{-5}$ | $n_1 = 103, n_2 = 120$<br>$p=2.0 \times 10^{-5}$ |
| *mat*<br>*swi6Δ bqt4Δ* | $n_1 = 77, n_2 = 75$<br>p=0.0002 | $n_1 = 100, n_2 = 59$<br>p=0.0459 |

DOI: https://doi.org/10.7554/eLife.32911.010

*Cech, 2001*; *Miyoshi et al., 2008*). In a *taz1+* background, Pot1-GFP foci localize to the NE in both early S and late S/G2 (*Figure 4E*) and become dislodged in late S/G2 upon *bqt4* deletion. In contrast, in a *bqt4Δ taz1Δ* setting in which telomere replication timing is deregulated, telomere dislodgement occurs throughout all stages monitored (*Figure 4E*). The more pronounced late S/G2-dependency of Taz1 delocalization (*Figure 4D*) versus Tel1L delocalization (*Figure 4C*) in the absence of Bqt4 likely reflects the distance separating Tel1L from the extreme chromosome terminus; Tel1L likely tends to replicate slightly earlier than the termini. Taz1 also promotes the passage of replication forks through the repetitive telomeric DNA (*Miller et al., 2006*); hence, while absence of Taz1 advances the initiation of telomere replication, it also hinders completion of replication (*Miller et al., 2006*). Consistently, telomeres remain dislodged throughout S/G2 phase in a *bqt4Δ taz1Δ* background.

Swi6 binds heterochromatic DNA at the *mat* locus, and causes early replication by promoting the recruitment of the replication initiation factor Sld3 (*Hayashi et al., 2009*). While Swi6 also binds telomeric heterochromatin, the replication timing of telomeres is set by Taz1 independently of heterochromatin/Swi6. The specific effect of Swi6 allowed us to examine the correlation between tethering of the *mat* locus and its replication status. In *swi6Δ* cells, *mat* remains tethered to the NE (*Figure 4F*), eliminating the possibility that Swi6 itself may affect tethering. However, in *swi6Δ bqt4Δ* cells, the *mat* locus is dislodged later, during late S/G2 phase (*Figure 4F*). Thus, Bqt4 is necessary for tethering of telomeres and the *mat* locus while these tethered loci are replicated.

The subnuclear positioning of replicating DNA has been visualized via tagged replication factors, such as the catalytic subunit of DNA polymerase α (Polα) (*Meister et al., 2007*; *Natsume et al., 2008*), which localizes to foci proposed to be the sites of active DNA replication. One or two such replication foci are observed at the NE during late S/G2 (*Meister et al., 2007*; *Natsume et al., 2008*). To examine whether telomeres colocalize with these late replication foci, we used cells expressing both Taz1-mCherry and Polα-GFP. *Figure 4G* shows a representative nucleus in late S/G2. Three Taz1 foci of similar intensity are visible in the shown z-plane; a fourth Taz1 focus is less intense as it represents an out-of-focus telomere in a different z-plane. Using the fluorescence background in the nucleus to approximate its boundary, we scored the position of the colocalized dots relative to the nuclear periphery (*Figure 4H*). Consistent with published data (*Meister et al., 2007*), we observe significant colocalization between telomeres and replication foci at the NE in *wt* cells. In contrast, while Taz1 and Polα show colocalization in the absence of Bqt4, this colocalization does not occur at the NE (*Figure 4H*), again indicating that the tendency of telomeres to replicate at the periphery requires Bqt4-mediated tethering.

## Bqt4 is required for resistance to DNA damaging agents during S-phase

The role of Bqt4 in localizing replicating telomeres and *mat* to the NE raises the question of whether such localization is important for their replication. To address this, we examined the effects of *bqt4+* deletion on DNA replication-related processes. Bqt4 is required for cellular resistance to agents that induce DNA damage during S-phase, as *bqt4Δ* cells are hypersensitive to both hydroxyurea (HU) and methyl methanesulfonate (MMS; *Figure 5A*); in contrast, Bqt4 is dispensable for resistance to bleomycin, which induces DNA damage independently of DNA replication (*Manolis et al., 2001*), and to the microtubule destabilizing agent thiabendazole (*Figure 5B*, *Figure 5—figure supplement 1A*). These results demonstrate a requirement for either Bqt4 per se and/or Bqt4-mediated NE tethering for robust DNA replication. To delineate these possibilities, we utilized strains harboring a deletion of the transmembrane domain of Bqt4 (Bqt4-ΔTM); Bqt4ΔTM has been shown to delocalize from the NE and diffuse throughout the nuclear interior (*Chikashige et al., 2009*). GFP-Bqt4-ΔTM confers similar MMS hypersensitivity to that of *bqt4+* deletion while GFP-Bqt4 alone does not (*Figure 5—figure supplement 1B*), indicating that the NE localization of Bqt4 is required for resistance to MMS.

We wondered if the DNA damage sensitivity of *bqt4Δ* cells stems from the heterochromatic nature of Bqt4-tethered regions, in which case the abolition of H3K9 methylation conferred by *clr4+* deletion would suppress the damage sensitivity. However, *clr4Δ* confers synthetic lethality with *bqt4Δ* on MMS (*Figure 5C*). Therefore, Bqt4 and Clr4 play separate, redundant roles in controlling MMS resistance. This suggests a scenario in which localization becomes particularly crucial when Bqt4-interacting chromosome regions are transcriptionally derepressed; enhanced transcription at

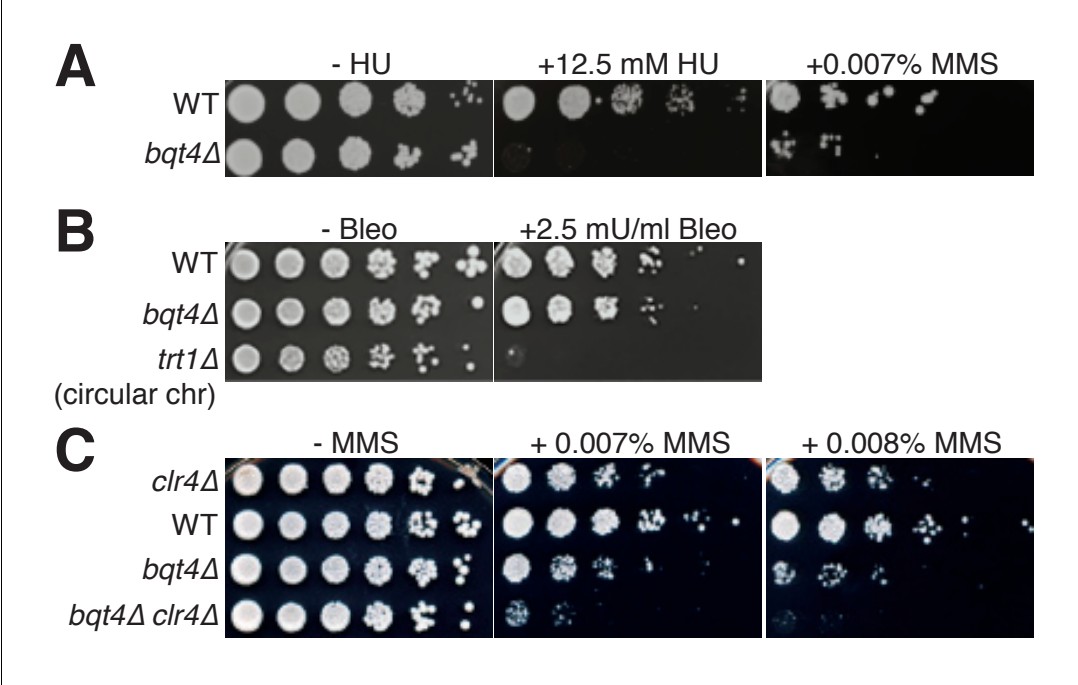

**Figure 5.** Bqt4 is required for resistance to agents that damage DNA during replication. (**A–C**) Five-fold serial dilutions of log-phase cultures of the indicated strains were stamped onto media containing HU, MMS, or Bleomycin (see Materials and methods). To control for the effectiveness of Bleomycin, we used telomerase-deficient survivals that have circular chromosomes, previously shown to be bleomycin hypersensitive (*Jain et al., 2010*).

DOI: https://doi.org/10.7554/eLife.32911.011

The following figure supplement is available for figure 5:

**Figure supplement 1.** The NE localization of Bqt4 is required for resistance to MMS, but not TBZ.

DOI: https://doi.org/10.7554/eLife.32911.012

these difficult-to-replicate regions may lead to excessive transcription/replication conflicts, causing severe MMS hypersensitivity.

## Bqt4 and Lem2 microenvironments regulate telomeric and centromeric heterochromatin

The hypersensitivity of *bqt4Δ* cells to agents that challenge DNA replication, along with the heterochromatic nature of the Bqt4-tethered chromosome regions, raised the question of whether heterochromatin is altered by Bqt4 loss. To assess this, we created strains in which chromatin was induced to position away from both the Lem2 and Bqt4 microdomains. Given our observation that Lem2 concentrates beneath the SPB while Man1 and Bqt4 occupy distinct and largely non-overlapping positions around the NE, we sought to move replicating chromatin to Man1-enriched regions. An ectopic tethering system was achieved by fusing the C-terminus of Man1 with the GFP-binding protein (GBP) (*Dodgson et al., 2013*), which was in turn fused to mCherry. Man1-GBP-mCherry localizes around the NE (*Figure 6A*), as does unmodified Man1. To tether replicating DNA to Man1-GBP-mCherry, we utilized Polα-GFP (*Damagnez et al., 1991*), which fully complements *wt* Pol α function (*Meister et al., 2007*) (*Figure 6—figure supplement 1*). In cells lacking GBP, Polα-GFP shows diffuse localization throughout the nucleus during early S-phase (*Figure 6B*). In contrast, Polα-GFP accumulates around the NE in S-phase cells harboring Man1-GBP (*Figure 6A*). To examine the location of active DNA replication, we visualized the single-strand DNA binding RPA (replication protein A) component Rad11, which localizes to replication forks (as well as DNA damage sites). In cells lacking GBP, multiple Rad11 foci are visible throughout the nucleus during S phase while Rad11 signal is diffuse in nonreplicating cells (*Figure 6—figure supplement 2A*) (*Zaaijer et al., 2016*). By contrast, in nearly all Man1-GBP/Pol α-GFP cells, the Rad11 foci colocalize with peripherally localized Polα-GFP (*Figure 6—figure supplement 2B*), suggesting that the bulk of DNA replication takes place at

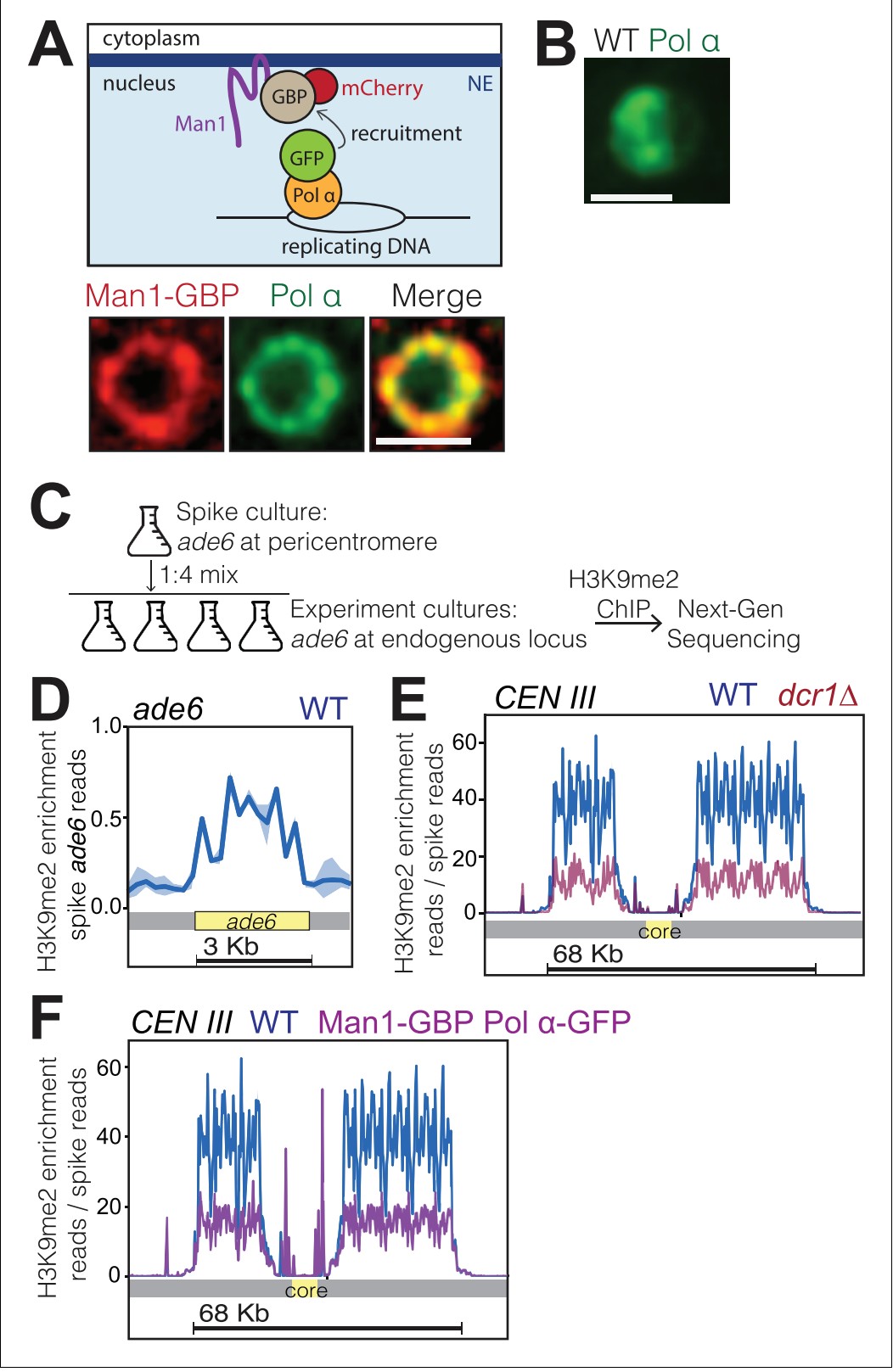

**Figure 6.** Bqt4-rich domains ensure faithful inheritance of heterochromatin. (**A**) Schematic of system designed to move replication forks to the NE. Coexpression of Polα-GFP and Man1-GBP recruits Polα-associated (replicating) chromatin to Man1-rich subdomains of the NE. Images below schematic show remobilization of Pol α-GFP to

*Figure 6 continued on next page*

*Figure 6 continued*

colocalize with Man1-GBP-mCherry encircling the nucleus. (**B**) In cells lacking Man1-GBP, Polα-GFP shows diffuse localization within the nuclear interior. (**C**) ChIP-seq experiments were performed as described in text and Methods, using H3K9Me2 antibody (ab1220, Abcam). (**D**) Ribbon line-plot of enrichment of H3K9Me over the *ade6 +* locus in three independent WT isolates. The solid blue line indicates median; width of the ribbon indicates range of enrichment levels among the three isolates. (**E–F**) Ribbon line-plot of enrichment of H3K9Me2 at centromere of Chr III. The central core region is indicated by the yellow box. The sharp peaks flanking the core region (**F**) align to small repetitive regions (~25 bp) present in multiple regions across the genome; it is therefore not possible to determine their source.

DOI: https://doi.org/10.7554/eLife.32911.013

The following figure supplements are available for figure 6:

**Figure supplement 1.** A C-terminal GFP tag on Polα does not interfere with centromeric silencing.

DOI: https://doi.org/10.7554/eLife.32911.014

**Figure supplement 2.** Man1-GBP pulls both Polα-GFP and replicating telomeres to Man1-rich domains.

DOI: https://doi.org/10.7554/eLife.32911.015

---

the Man1 microdomains; Taz1-Man1 colocalization can also be seen to increase in a Man1-GBP background when Polα-GFP is introduced (*Figure 6—figure supplement 2C–E*), suggesting the telomere replication is moved from Bqt4 microdomains to Man1-domains. Hence, this approach enabled us investigate the effect of inducing the bulk of DNA replication to occur at the periphery at sites distinct from those rich in Lem2 and Bqt4.

To detect changes in heterochromatin enriched with H3K9me2, we used a modified ChIP-Seq protocol that entails spiking every sample with chromatin from a strain in which the endogenous *ade6+* gene has been deleted and ectopically inserted within the pericentric heterochromatin of centromere I. As all the experimental strains used contain only the single endogenous copy of *ade6 +*, which lies in a euchromatic region (*Cam et al., 2005*), the signal emanating from ectopic *ade6+* serves as a constant reference. This approach for normalization of H3K9me2 enrichment levels is similar to the previously described ChIP-Rx method (*Orlando et al., 2014*). We mixed a liquid culture of the reference (spike) strain at 1:4 ratio with each sample of the experimental cultures (*Figure 6C*). The resulting spiked samples were then cross-linked and processed for ChIP. *Figure 6D* shows the enrichment of H3K9me2 over the *ade6* open reading frame, confirming our ability to detect the spike reads at the indicated spike ratio. To validate this normalization approach, we compared three independent *wt* strains with three independent *dcr1Δ* strains known to have vastly reduced levels of pericentric H3K9me2 (*Hall et al., 2002*; *Volpe et al., 2002*). After normalizing to the reads obtained at the *ade6* locus (no normalization to input), the number of reads covering the heterochromatic pericentromeres showed no significant variation among the *wt* strains (*Figure 6E*). In contrast, the strains lacking Dicer (*Hall et al., 2002*; *Volpe et al., 2002*) are similar to each other but show substantially reduced read coverage relative to *wt* at the pericentromeric regions (*Figure 6E*). Hence, normalization to the spike is sufficient to eliminate technical variations introduced across multiple samples.

Cells harboring Man1-GBP and Polα-GFP were subjected to ChIP-Seq as described above. The read coverage for all pericentromeric regions was substantially reduced in Man1-GBP Polα-GFP cells relative to *wt* (*Figure 6F*). Hence, centromeric heterochromatin maintenance is compromised by inducing centromeres to replicate at Man1 regions, away from the Lem2 microdomains. In addition to reinforcing previous observations of a role for Lem2 in pericentric heterochromatin maintenance, these data are consistent with our observation that Man1 microdomains are spatially distinct from Lem2 microdomains.

To assess the effect of NE-proximal replication on subtelomeric heterochromatin, we performed ChIP-Seq on three independent *bqt4Δ* isolates. In a *wt* background, the 35 Kb region near the left end of Chr I is enriched with H3K9me2 (*Figure 7A*). In the absence of Bqt4, H3K9me2 levels are increased over this subtelomeric region (*Figure 7A*); subtelomere 2L also shows increased H3K9me2 enrichment in the *bqt4Δ* setting (*Figure 7B*). Moreover, while a sharp decrease in H3K9me2 enrichment is noticeable at the boundary between telomeric heterochromatin and the euchromatic chromosome arm in *wt* cells, this distinct H3K9me2 transition zone is not observed in *bqt4Δ* cells, in which additional H3K9me2 peaks extend into adjacent, normally euchromatic, regions. Telomere

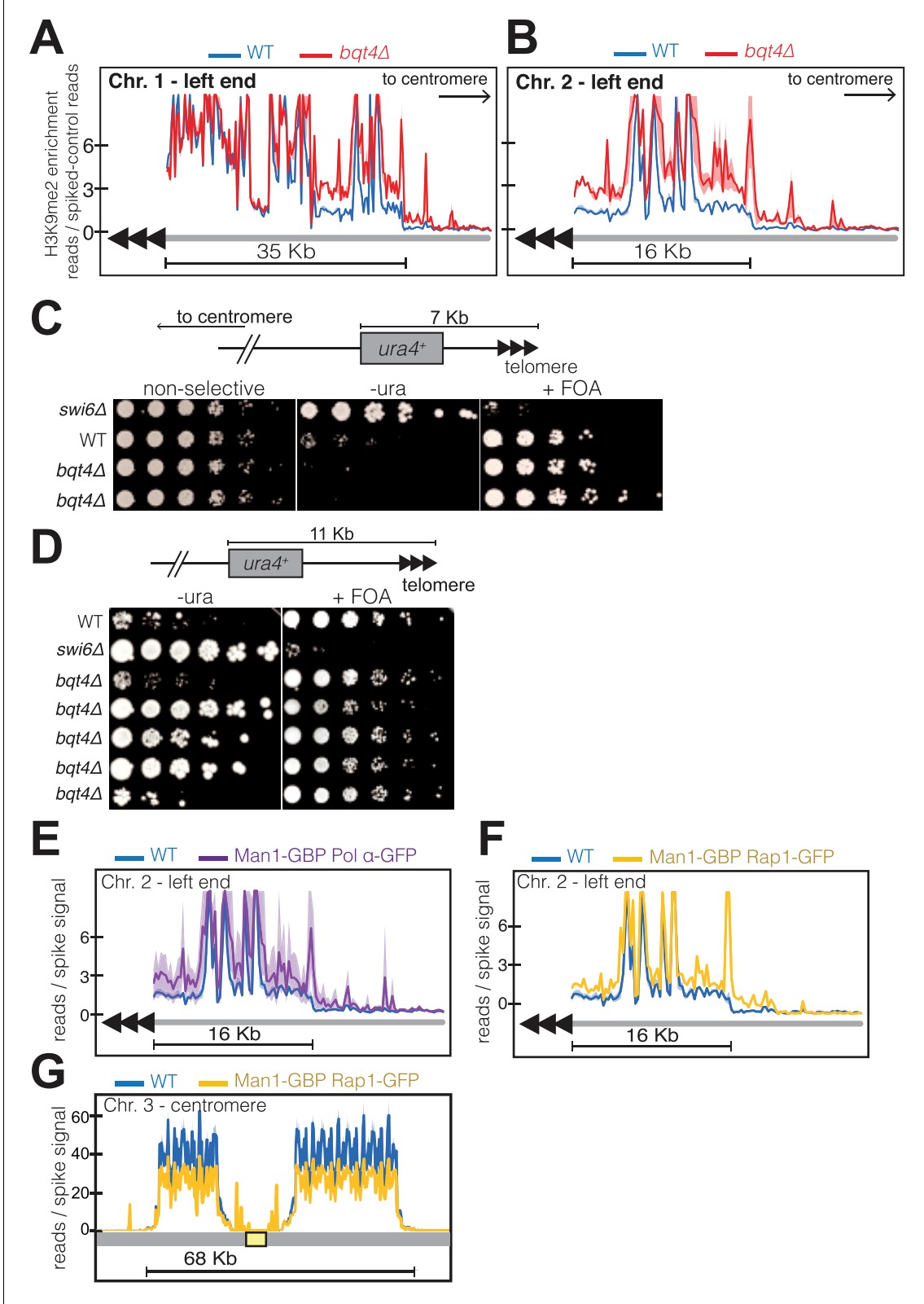

**Figure 7.** Bqt4 ensures proper heterochromatin maintenance at subtelomeric regions. (**A–B**) ChIP-seq experiments (H3K9Me2) with three independent isolates of each genotype. Plot annotation as described in **Figure 6D**. (**C**) Five-fold dilutions of cells harboring the indicated genotypes and *ura4+* gene inserted~7 kb from the right end of chromosome II. The two *bqt4Δ* rows contain cultures of two independent isolates. (**D**) Experiment performed as

*Figure 7 continued on next page*

*Figure 7 continued*

described in C, with five independent *bqt4Δ* isolates. (**E–F**) ChIP-Seq data covering the left end of Chr II is plotted as described in *Figure 6D*. (**G**) ChIP-seq data for centromeric region of Chr III.

DOI: https://doi.org/10.7554/eLife.32911.016

The following figure supplements are available for figure 7:

**Figure supplement 1.** Bqt4 regulates H3K9Me2 levels at pericentric and subtelomeric regions but not the *mei4+* locus.

DOI: https://doi.org/10.7554/eLife.32911.017

**Figure supplement 2.** Bqt4 affects silencing at mat but not the distal subtelomere.

DOI: https://doi.org/10.7554/eLife.32911.018

proximal regions at other chromosome ends also display defective H3K9me2 organization (*Figure 7—figure supplement 1A*). However, non-telomeric heterochromatic regions, such as that surrounding the *mei4* gene (1.4 Mb from Tel2L) (*Zofall et al., 2012*), show normal H3K9me2 profiles in the absence of Bqt4 (*Figure 7—figure supplement 1B*). In contrast to subtelomeres, pericentric H3K9me2 enrichment is slightly reduced by *bqt4+* deletion (*Figure 7—figure supplement 1A*) and H3K9me2 enrichment at the *mat* locus is substantially reduced (*Figure 7—figure supplement 2A*).

To probe our ChIP-seq observations with a physiological assay, we utilized a strain harboring a *ura4* reporter gene inserted 7 Kb from the chromosome end (*Figure 7C*). This ectopic *ura4* gene is known to be transcriptionally repressed and bound by Swi6 in *wt* cells (*Kanoh et al., 2005*), conferring poor growth on media lacking uracil (-ura). The same *wt* strain shows robust growth on media containing uracil and 5-Fluoroorotic acid (FOA), which is toxic to cells expressing *ura4*. This is largely true in a *bqt4Δ* setting as well, while cells lacking Swi6 show the expected derepression of *ura4*. However, a subtle but noticeable difference emerges when comparing the growth of *wt* and *bqt4Δ* strains on -ura; the *bqt4Δ* strains grow more poorly, indicating stronger repression of *ura4* expression. In a similar experiment using cells with *ura4* inserted 11 Kb from the chromosome end (*Figure 7D*), we found *ura4* repression, consistent with previous observations (*Kanoh et al., 2005*). However, substantial growth variability emerged among five *bqt4Δ* isolates, with one showing similar growth to *wt* (poor on -ura and robust on FOA) and three *bqt4Δ* isolates (*Figure 7D*, rows 4–6) showing robust growth on both -ura and FOA, indicating enhanced variegation. Hence, the more centromere-proximal *ura4* reporter is repressed in the absence of Bqt4, but can be derepressed under selection for *ura4* prototrophy. Therefore, Bqt4 is dispensable for subtelomeric silencing, but the extent and robustness of subtelomeric silencing are both limited by the presence of Bqt4. In contrast to the subtelomeres, Bqt4 does not appear to regulate silencing at the telomeric regions (~500 bp from chromosome end; *Figure 7—figure supplement 2B*).

To determine whether the effect of Bqt4 on telomeric heterochromatin is mediated by tethering to the Bqt4-adjacent microenvironment, we examined H3K9me2 enrichment at telomeres in the induced tethering strains harboring Man1-GBP and Polα-GFP. In these strains, Bqt4 is present but DNA replication is induced to take place away from the Bqt4 microdomain. At Tel2L, the amplitudes of H3K9me2 peaks are elevated in the' induced tethering' background relative to *wt* (*Figure 7E*); this held true for other subtelomeres as well. Furthermore, the boundary regions appear to be affected in a manner resembling that of the *bqt4Δ* setting. Therefore, telomere replication in Man1-adjacent, rather than Bqt4-adjacent, microdomains leads to mis-regulated H3K9me2 maintenance.

As induced replication near Man1 affects both subtelomeric and centromeric H3K9me2, we sought to address the possibility that decreased levels of centromeric H3K9me2 result from the increase at subtelomeres. To tether telomeric regions to Man1 without altering pericentric localization, we performed ChIP-seq in cells expressing Rap1-GFP and Man1-GBP; in this scenario, the Rap1-bound telomeres will localize to Man1 domains while centromere replication is unaffected. In Man1-GBP/Rap1-GFP cells, subtelomeric H3K9me2 increases in Man1-GBP/Rap1-GFP cells to an extent similar to that seen in Man1-GBP/Polα-GFP cells (*Figure 7F*). Centromeric H3K9me2 peak amplitudes are modestly reduced in this scenario, but this reduction is minor compared with the reduction seen in Man1-GBP/Polα-GFP scenarios (compare *Figure 6F* with *Figure 7G*). Hence, we suspect that while titration effects due to increased subtelomeric H3K9me2 levels contribute modestly to the pericentric effect of inducing replication near Man1, the Bqt4-microdomain that is at least partially vacated in the Man1-GBP/Polα-GFP scenario regulates pericentric replication directly.

## Lem2 and Bqt4 become crucial for viability in the absence of Dcr1

The foregoing results outline heterochromatin replication and assembly functions of Lem2- and Bqt4-adjacent microdomains, but also demonstrate that Bqt4 regulates Lem2 itself. To gain insight into the mechanisms underlying Lem2- and Bqt4-mediated chromatin regulation, we examined a series of genetic interactions by constructing doubly and triply mutated strains.

The *lem2Δ* and *dcr1Δ* single mutant cells show moderately (~5 fold) reduced growth relative to *wt* (*Figure 8A*). Loss of the anti-silencing factor Epe1 restores *wt* levels of viability to *dcr1Δ* but not *lem2Δ* cells, indicating that Lem2 acts in a pathway independent of heterochromatin maintenance to promote viability. Moreover, the double deletion *lem2Δ dcr1Δ* shows markedly reduced viability, demonstrating that Lem2 acts in a pathway separate from (and redundant with) Dicer-mediated pathways to promote viability. Accordingly, deletion of *epe1* confers suppression of the *lem2Δ dcr1Δ* inviability, but not to the level of an *epe1Δ dcr1Δ* strain (*Figure 8A*). As discussed below, these results, along with our ChIP-seq and localization data, suggest that Lem2 functions both in pericentric heterochromatin maintenance and in centromere/kinetochore maintenance, which when compromised will make Dicer essential (*Folco et al., 2008*; *Klutstein et al., 2015*). Consistent with this idea, we find that we cannot obtain cells harboring *lem2+* deletion along with a temperature sensitive allele of Cenp-A (*cnp1-1*) at any temperature (*Takahashi et al., 2000*). Overexpression of the Cenp-A loading factor Ams2 has been shown to rescue the temperature sensitivity of *cnp1-1* (*Figure 8B*) (*Chen et al., 2003*), and introduction of overexpressed Ams2 does confer limited (though vastly reduced relative to *lem2+*) viability to *lem2Δ cnp1-1* cells at permissive temperature (32°C, *Figure 8B*). However, *lem2+* deletion abolishes suppression of *cnp1-1* temperature sensitivity by overexpressed Ams2 (*Figure 8B*).

In contrast to Lem2 or Dcr1, the loss of Bqt4 alone does not compromise growth under unperturbed conditions (*Figure 8C*). However, *bqt4Δ* and *dcr1Δ* are synthetically sick and *bqt4Δ lem2Δ* cells are inviable (data not shown) (*Tange et al., 2016*). Hence, in the absence of Lem2 or Dcr1, Bqt4 becomes crucial for viability. Below we propose a model in which Bqt4 defines domains that confer viability in the face of transcription/replication collisions (which are exacerbated in *dcr1Δ*; (*Kloc et al., 2008*; *Zaratiegui et al., 2011*) while Lem2 plays roles in kinetochore maintenance. Dicer intersects with both of these processes by promoting RNAP2 dislodgement as well as heterochromatin formation.

## Discussion

Here we delineate biochemically distinct microenvironments at the nuclear periphery, with distinct capacities to regulate centromeric and telomeric chromatin replication (*Figure 9*). In addition to the Lem2-rich domain beneath the SPB, these microenvironments comprise Man1-rich domains and separate Bqt4-rich domains. Bqt4 regulates the replication of specific chromosome regions *via* two separable modes: First, by tethering specific heterochromatic regions, Bqt4-microdomains promote the fidelity of DNA replication and reassembly of heterochromatin in the wake of the replication fork. Second, Bqt4 mobilizes Lem2 to sites around the NE to promote pericentric heterochromatin maintenance. Our results suggest separable roles for Lem2 as well; the subset of Lem2 molecules sequestered beneath the SPB influences CenpA loading and kinetochore maintenance, while the Bqt4-mobilized subset of Lem2 molecules promotes pericentric heterochromatin maintenance. Collectively, our data suggest a model in which the central function of Bqt4 is to provide a 'safe zone' within the nucleus in which duplication of difficult-to-replicate repeats, as well as their underlying chromatin, is facilitated.

Although Bqt4 has been known to tether telomeres to the NE, we find that this tethering role is specific to telomeres undergoing replication; redundant pathways are clearly capable of tethering nonreplicating telomeres to the NE. Conceivably, the relatively high mobility of Bqt4 along the NE allows the flexibility to keep contact with telomeres as they unwind and accommodate the replisome. As Bqt4 also tethers the *mat* locus while it is replicating, we infer that multiple loci, perhaps all of them repetitive and/or heterochromatic, are regulated in this manner. Nevertheless, it is clear that not all loci come under such Bqt4 control, as the euchromatic arm locus we studied (*cut3+*) shows neither NE enrichment nor Bqt4 dependent localization. The specificity of Bqt4 interaction with non-telomeric loci may be determined *via* its N-terminal DNA-binding domain, which shows

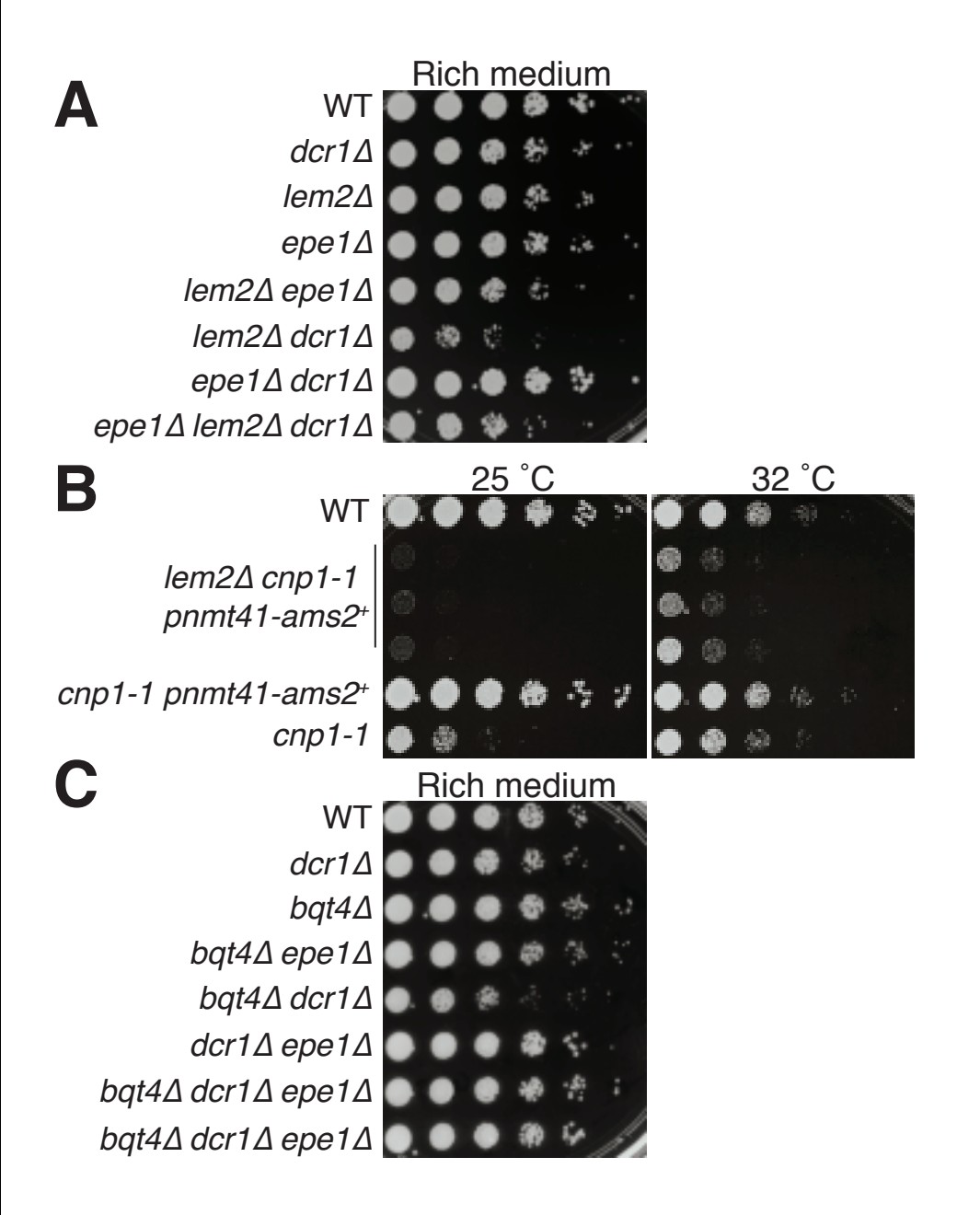

**Figure 8.** Genetic analysis shows roles of Bqt4 and Lem2 in promoting cellular viability. (**A**) Five-fold serially diluted cultures of the indicated strains were plated on rich media (YE5S) at 32°C. (**B**) Growth of cells harboring a temperature sensitive allele of *cnp1+* (*cnp1-1*) with and without overexpressed (*nmt41*-controlled) Ams2 was assessed with five-fold serially diluted cultures incubated at 25°C and 32°C. (**C**) Viability assay was performed as described in (**A**).

DOI: https://doi.org/10.7554/eLife.32911.019

homology to a family of fungal transcription factors; alternatively, by analogy with the telomeric Taz1-Rap1-Bqt4 interaction, Bqt4 may contact *mat via* other DNA binding proteins.

Bqt4-microdomains are important for cellular resistance to agents that cause DNA damage during replication. We propose a model in which this sensitivity stems from damage specifically to the chromatin regions (eg, telomeres) that are tethered by Bqt4 to the NE. Indeed, these repetitive

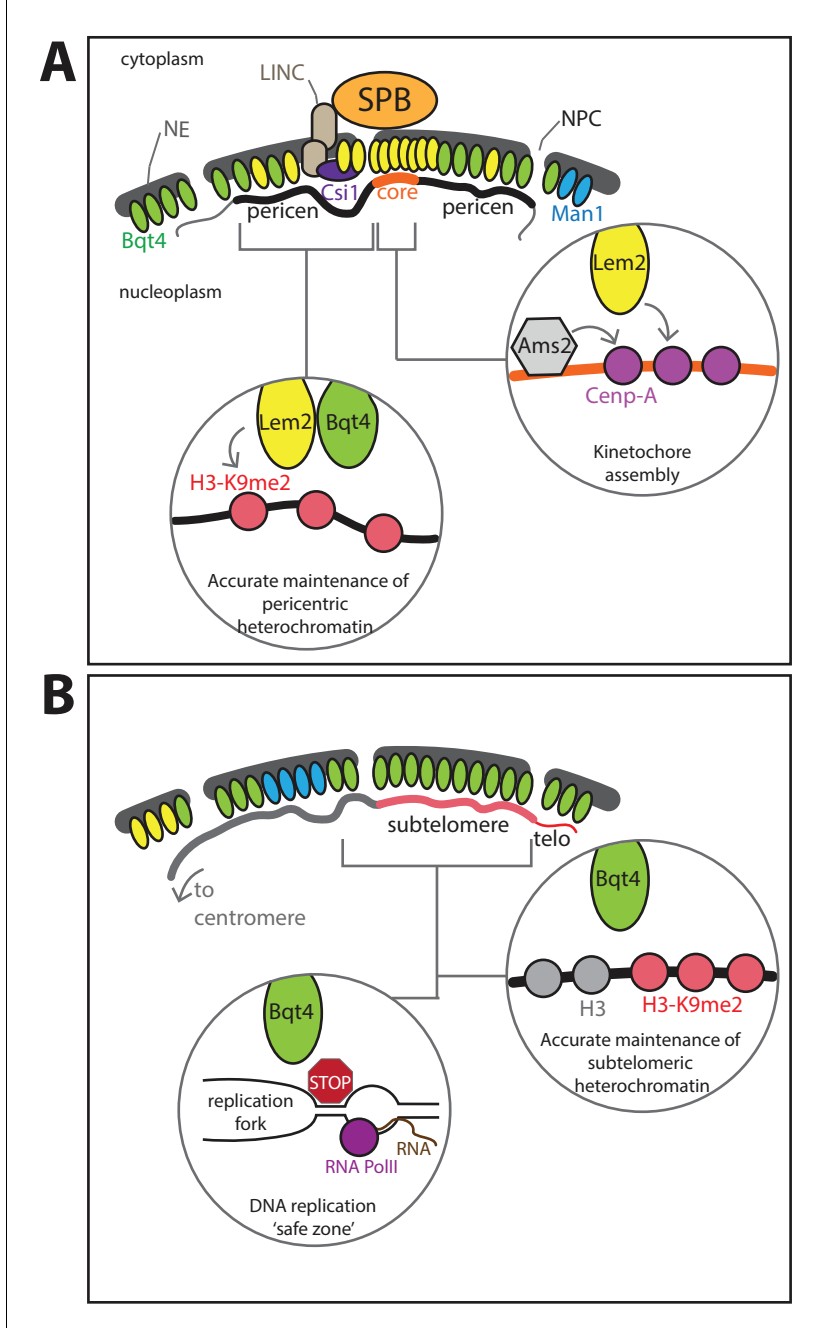

**Figure 9.** Model for roles of Lem2- and Bqt4-microdomains. (**A**) Lem2 (yellow ovals) exists in two populations, the most concentrated fraction localizing beneath the SPB, where Lem2 is stabilized by Csi1. This Lem2 fraction promotes kinetochore maintenance (right insert) at the central core of the centromere (core). Lem2 is also mobilized by Bqt4 (green ovals) to sites away from the SPB (left inset), where it functions in pericentric replication and silencing. (**B**) Bqt4 localizes to distinct sites around the NE, some of which overlap with Lem2 and some of which do not; Bqt4 rarely if ever overlaps with Man1 (blue ovals). Bqt4-microdomains are 'safe zones' in which repetitive heterochromatic sequences, pericentromeres (in the Lem2-overlapping regions), telomeres and *mat* locus (in the non-Lem2 overlapping regions), can be replicated without suffering excessive replication/transcription collisions, perhaps by excluding RNAP2 or concentrating Pfh1 helicase (red stop sign, left inset); these safe zones also promote the high fidelity maintenance of pericentric, subtelomeric and *mat* heterochromatin (right inset).
DOI: https://doi.org/10.7554/eLife.32911.020

regions are particularly challenging for the replication machinery (*Miller et al., 2006*), and HU or MMS likely exacerbate the level of stalled and/or collapsed forks in these regions. In particular, we suggest that Bqt4-rich domains are specialized microenvironments in which collisions between replication and transcription machineries are robustly handled or avoided. Such collisions are a known danger at the telomere-adjacent rDNA regions, and become more likely at all regions undergoing high levels of transcription and/or stalled replication. The synthetic lethality of *bqt4+* and *clr4+* on MMS suggests that when H3K9me-containing heterochromatin is abolished, the resulting transcriptional increase heightens the danger of collisions with the replisome, in turn heightening the need for Bqt4-mediated tethering. Consistently, Bqt4 becomes crucial for viability even in unperturbed conditions when Dicer, which has been shown to promote transcriptional termination at sites of replication stress *via* eviction of RNAP2, is missing. The notion that Bqt4-microdomains are 'safe zones' for replication of repetitive regions can explain our initial observation that Bqt4 becomes important in HAATI cells, in which rDNA repeats form all chromosomal termini, leading cells to harbor on average six-fold more rDNA than *wt* cells.

The properties of Bqt4-rich regions that make them suitable for averting replication/transcription collisions remain to be determined. These regions may exclude RNAP2 or concentrate the Pif1 helicase, which allows replication to negotiate the transcription machinery (*Sabouri et al., 2012*). The accumulation of telomeric transcripts (TERRA) is reduced specifically during telomere replication (*Azzalin and Lingner, 2015*); conceivably, this reduction depends on localization to the Bqt4-rich domain.

Our microscopy shows distinct localization patterns for centromere cores and pericentric regions; centromeric cores colocalize with Lem2 beneath the SPB while pericentromeres localize up to a micron away, potentially colocalizing with non-SPB-adjacent, Bqt4-dependent Lem2 sites. Hence, the effect of Bqt4 on pericentric heterochromatin maintenance likely stems from loss of Lem2 from these SPB-distal sites. However, as Bqt4 and Lem2 are synthetically lethal (data not shown and (*Tange et al., 2016*), they are clearly required for additional pathways uniquely regulated by either protein. The strong synthetic interactions we observe between *lem2Δ* and *cnp1-1* suggest that Lem2 regulates Cnp1 maintenance at the centromeric core. Compromised Cnp1 maintenance would necessitate robust pericentric heterochromatin, which is required for establishment of Cnp1 at centromere sequences (*Folco et al., 2008*); this establishment function is likely compromised when the replication 'safe zone' conferred by Bqt4 is lost. The potential for replication/transcription collision is severe at pericentromeres, where RNAi-mediated release of RNAP2 is required to prevent stalled forks and thereby promote heterochromatin inheritance (*Kloc et al., 2008*; *Zaratiegui et al., 2011*). Hence, we propose that in a *lem2Δ dcr1Δ* double mutant, a 'perfect storm' of factors severely confounds viability; Cnp1 maintenance is compromised, replication of the region is highly problematic, and pericentric heterochromatin is abolished, disallowing both kinetochore establishment and kinetochore maintenance. Such considerations may also underlie the complete inviability of *lem2Δbqt4Δ* cells (data not shown) (*Tange et al., 2016*).

Similar replication challenges may alter heterochromatin reassembly at *bqt4Δ* subtelomeres; furthermore, by releasing limiting heterochromatin factors, compromised pericentric heterochromatin may contribute to the increased H3K9me2 levels at *bqt4Δ* subtelomeres (*Tadeo et al., 2013*). We note that the effects of *bqt4+* deletion on pericentric and subtelomeric H3K9me2 levels are subtle. Likewise, Bqt4 loss fails to confer appreciable sensitivity to the microtubule depolymerizing agent thiabendazole (TBZ, *Figure 5—figure supplement 1A*) in contrast with Dcr1 loss, which confers TBZ hypersensitivity due to loss of pericentric heterochromatin (*Provost et al., 2002*). Hence, while pericentric heterochromatin levels are reduced by loss of Bqt4, they are high enough to prevent pericentric cohesin levels from dropping beneath the critical threshold that governs TBZ sensitivity.

While our work has revealed roles of a specific pair of microdomains in regulating distinct chromatin regions, we hypothesize that this reflects a wider role for distinct NE subdomains in influencing chromatin structure and function at other loci. The orderly formation of nuclear microenvironments is even more critical in mammalian genomes with their higher proportion of repetitive sequences and diverse range of chromatin types found along the length of a chromosome (*Akhtar et al., 2013*; *Gerhardt et al., 2016*; *Simonis et al., 2006*; *Zullo et al., 2012*). For difficult-to-replicate chromatin regions, the presence of distinct nuclear subdomains, with specific molecular compositions tailored to interactions with distinct chromatin subtypes, may coordinate the fidelity of

replication of each subtype, ensuring that entire chromosomes are inherited with both genetic and epigenetic information intact.

# Materials and methods

## Key resources table

| Reagent type (species) or resource | Designation | Source | Identifiers | Additional information |
|---|---|---|---|---|
| Chemical compound | Thiabenzadole | Sigma-Aldrich | Cat#T8904 | |
| Chemical compound | Methyl methanesulfonate | Sigma-Aldrich | Cat#66-27-3 | |
| Chemical compound | 5-Fluoroorotic Acid | US Biological | Cat#207291-8-4 | |
| Chemical compound | nourseothricin dihydrogen sulfate | US Biological | Cat#N5374-74 | |
| Chemical compound | G418 sulfate (Geneticin) | Invitrogen | Cat#11811031 | |
| Chemical compound | Hygromycin B | Invitrogen | Cat#10687010 | |
| Chemical compound | Hydroxyurea | Sigma-Aldrich | Cat#H8627 | |
| Reagent for microscopy | glass-bottom dish | MatTek | Cat#P35G-1.5–14 C | |
| Reagent for microscopy | GFP booster_ATTO488 | Chromotek | Cat#gba488-100 | |
| Reagent for microscopy | RFP booster_ATTO594 | Chromotek | Cat#rba594-100 | |
| Reagent for microscopy | Vectashield | Vector Labs | Cat#H-1000 | |
| Reagent for microscopy | 16% Paraformaldehyde aqueous | Electron Microscopy Sciences | Cat#15710 | |
| Reagent for microscopy | 8% glutaraldehyde | Electron Microscopy Sciences | Cat#16019 | |
| Reagent for microscopy | Zymolyase-100T | MP Biomedicals | Cat#320932 | |
| Reagent for ChIP | anti-H3K9Me2 ChIP Grade | Abcam | Cat#mAbcam1220 | |
| Reagent for ChIP | Complete, EDTA-free Protease inhibitor cocktail (Roche) | Sigma-Aldrich | Cat#11873580001 | |
| Reagent for ChIP | Phenylmethylsulfonyl fluoride | Sigma-Aldrich | Cat#P7626 | |
| Reagent for ChIP | Zirconium beads, 0.7 mm diameter | BioSpec Products | Cat#11079107zx | |
| Reagent for ChIP | Dynabeads Protein G | Invitrogen | Cat#10003D | |
| Reagent for ChIP | ChIP Elute Kit | Clontech | Cat#634887 | |
| Reagent for ChIP | milliTUBE 1 ml AFA Fiber | Covaris | Cat#520130 | |
| Reagent for ChIP | DNA SMART ChIP-Seq Kit | Clontech | Cat#634865 | |
| Reagent for ChIP | Agencourt AMPure XP | Beckman Coulter | Cat#A63880 | |
| Reagent for ChIP | NEBNext Library Quant Kit | New England BioLabs | Cat#E7630S | |
| Strain (*Schizosaccharomyces pombe*) | *h⁺ ade6 M216 his3-D1 leu1-32 ura4-D18* | Lab stock | JCF109 | |
| Strain (*Schizosaccharomyces pombe*) | *h⁻ ade6-M210 leu1-32 ura4-D18 lys1+::GFP-bqt4* | This study | JCF11923 | Plasmid pCSS18 (*Chikashige et al., 2009*) was integrated at lys1 locus. |
| Strain (*Schizosaccharomyces pombe*) | *h90 man1-GFP-kanMX6 aur1R-mCherry-bqt4 bqt4Δ: : natMX6 leu1-32 ura4-D18* | This study | JCF1266 | |

*Continued on next page*

*Continued*

| Reagent type (species) or resource | Designation | Source | Identifiers | Additional information |
|---|---|---|---|---|
| Strain (*Schizosaccharomyces pombe*) | h90 lem2-GFP-kanMX6 aur1R-mCherry-bqt4 bqt4Δ: : natMX6 ura4-D18 | This study | JCF1268 | |
| Strain (*Schizosaccharomyces pombe*) | lem2-GFP-kanMX6 man1-td Tomato-hygMX6 leu1-32 ura4-D18 | This study | JCF18885 | |
| Strain (*Schizosaccharomyces pombe*) | lem2-GFP-kanMX6 hht1-mRFP-hygMX6 | This study | JCF12191 | |
| Strain (*Schizosaccharomyces pombe*) | lem2-GFP-kanMX6 hht1-mRFP-hygMX6 bqt4::natMX6 | This study | JCF12201 | |
| Strain (*Schizosaccharomyces pombe*) | ura4-D18 lem2-GFP-KanMX6 | This study | JCF12148 | |
| Strain (*Schizosaccharomyces pombe*) | ura4-D18 lem2-GFP-KanMX6 bqt4::nat | This study | JCF12157 | |
| Strain (*Schizosaccharomyces pombe*) | lem2-GFP-kanMX6 sid4-mCherry-natMX6 aur1R-Pnda3-mCherry-atb2 | This study | JCF14537 | |
| Strain (*Schizosaccharomyces pombe*) | lem2-GFP-kanMX6 sid4-mCherry-natMX6 aur1R-Pnda3-mCherry-atb2 csi1::his3+ | This study | JCF14540 | |
| Strain (*Schizosaccharomyces pombe*) | lem2-GFP-kanMX6 bqt4::natMX6 csi1::ura4+ | This study | JCF14573 | |
| Strain (*Schizosaccharomyces pombe*) | lem2-GFP-kanMX6 taz1-mCherry-natMX6 csi1::his3+ | This study | JCF14535 | |
| Strain (*Schizosaccharomyces pombe*) | h⁻ taz1-mCherry-natMX6 lys1+ ::GFP-bqt4 | This study | JCF14578 | |
| Strain (*Schizosaccharomyces pombe*) | h⁻ taz1-mCherry-natMX6 man1-GFP-KanMX6 | This study | JCF14565 | |
| Strain (*Schizosaccharomyces pombe*) | h⁺ taz1-mCherry-natMX6 lem2-GFP-KanMX6 | This study | JCF14532 | |
| Strain (*Schizosaccharomyces pombe*) | h⁺ mis6-mCherry-hygMX6 lem2-GFP-KanMX6 | This study | JCF14543 | |
| Strain (*Schizosaccharomyces pombe*) | h⁺ mis6-mCherry-hygMX6 man1-GFP-KanMX6 | This study | JCF14533 | |
| Strain (*Schizosaccharomyces pombe*) | ade6-210 leu1-32 lys1-131 ura4-D18 sod2::kanr-ura4+-lacOp his7+::lacI-GFP cut11-RFP-hygMX6 | This study | JCF10756 | |
| Strain (*Schizosaccharomyces pombe*) | ade6-210 leu1-32 lys1 + ura4-D18 sod2::kanr-ura4+-lacOp his7+::lacI-GFP cut11-RFP-hyg bqt4::nat | This study | JCF10771 | |
| Strain (*Schizosaccharomyces pombe*) | ade6-210 leu1-32 lys1-131 ura4-D18 sod2::kanr-ura4+-lacOp his7+::lacI-GFP cut11-RFP-hyg lem2::nat | This study | JCF12202 | |
| Strain (*Schizosaccharomyces pombe*) | his2::kanMX6-ura4+-lacOp his7+::lacI-GFP cut11-RFP-hygMX6 | This study | JCF10732 | |
| Strain (*Schizosaccharomyces pombe*) | his2::kanMX6-ura4+-lacOp his7+::lacI-GFP cut11-RFP-hyg MX6 bqt4::natMX6 | This study | JCF10740 | |
| Strain (*Schizosaccharomyces pombe*) | cut3+::lacOp his7+::lacI-GFP cut11-RFP-hygMX6 | This study | JCF10755 | |
| Strain (*Schizosaccharomyces pombe*) | cut3+::lacOp his7+::lacI-GFP cut11-RFP-hygMX6 bqt4::nat | This study | JCF11930 | |
| Strain (*Schizosaccharomyces pombe*) | taz1-mCherry-nat pLT1-2 (nmt1:lsh1-GFP-LEU2) | This study | JCF10701 | |

*Continued on next page*

Continued

| Reagent type (species) or resource | Designation | Source | Identifiers | Additional information |
|---|---|---|---|---|
| Strain (*Schizosaccharomyces pombe*) | taz1-mCherry-nat bqt4::hygMX6 pLT1-2 (nmt1:Ish1-GFP-LEU2) | This study | JCF10723 | |
| Strain (*Schizosaccharomyces pombe*) | pot1-mRFP:kanMX6 cut11-3xPK-GFP:ura4 + leu1-32 ura4-D18 (WT) | Lab stock | JCF3649 | |
| Strain (*Schizosaccharomyces pombe*) | cut11-RFP-hyg tpz1-GFP-Kan taz1::ura4+ | This study | JCF12293 | |
| Strain (*Schizosaccharomyces pombe*) | cut11-RFP-hyg tpz1-GFP-Kan bqt4::nat | This study | JCF12291 | |
| Strain (*Schizosaccharomyces pombe*) | cut11-RFP-hygMX6 tpz1-GFP :kanMX6 bqt4::nat taz1::ura4+ | This study | JCF12290 | |
| Strain (*Schizosaccharomyces pombe*) | his2::kanMX6-ura4+-lacOp his7+::lacI-GFP cut11-RFP-hyg bqt4::nat swi6::leu2+ | This study | JCF10807 | |
| Strain (*Schizosaccharomyces pombe*) | his2::kanMX6-ura4+-lacOp his7+::lacI-GFP cut11-RFP-hyg swi6::leu2+ | This study | JCF 12279 | |
| Strain (*Schizosaccharomyces pombe*) | h⁻ swi7-GFP::kanMX6 taz1-mCherry-natMX6 | This study | JCF11905 | |
| Strain (*Schizosaccharomyces pombe*) | swi7-GFP::kanMX6 taz1-mCherry-nat bqt4::hygMX6 | This study | JCF14504 | |
| Strain (*Schizosaccharomyces pombe*) | h⁻ bqt4::nat ade6-M210 his3-D1 leu1-32 ura4-D18 | This study | JCF11933 | |
| Strain (*Schizosaccharomyces pombe*) | h⁺ ade6 M216 trt1::hygMX6 | *Jain et al. (2010)* | JCF6858 | Circular survivor |
| Strain (*Schizosaccharomyces pombe*) | h⁺ ade6 M210 clr4::kanMX6 | Lab stock | JCF3113 | |
| Strain (*Schizosaccharomyces pombe*) | h⁻ clr4::kanMX6 bqt4::hygMX6 | This study | JCF10795 | |
| Strain (*Schizosaccharomyces pombe*) | h⁺ swi7-GFP-kanMX6 man1-GBP-mCherry-hph | This study | JCF11950 | GBP plasmids were gifts from Masamitsu Sato, University of Tokyo |
| Strain (*Schizosaccharomyces pombe*) | h⁻ swi7-GFP-kanMX6 | *Natsume et al. (2008)* | JCF10835 | |
| Strain (*Schizosaccharomyces pombe*) | h⁺ swi7-GFP-kanMX6 man1-GBP-mCherry-hph bqt4::naMX6 | This study | JCF14517 | |
| Strain (*Schizosaccharomyces pombe*) | h⁹⁰ otr1R::ade6 tel1L::his3 tel2L::ura4 ade6-210 his3-D1 leu1-32 ura4-D18 | *Nimmo et al., 1998* | JCF6712 | Spike strain in ChIP-seq experiments. Used for telomere silencing assay |
| Strain (*Schizosaccharomyces pombe*) | h⁻ dcr1::ura4+ | This study | JCF12179 | |
| Strain (*Schizosaccharomyces pombe*) | h⁺ ade6 M216 his3-D1 leu1-32 ura4-D18 bqt4::natMX6 | This study | JCF14512 | |
| Strain (*Schizosaccharomyces pombe*) | h⁻ leu1-32 ura4-D18 T2R1-4137::ura4+ | *Kanoh et al. (2005)* | JCF10833 | |
| Strain (*Schizosaccharomyces pombe*) | h⁻ swi6::LEU2 leu1-32 ura4-D18 T2R1-4137::ura4+ | *Kanoh et al. (2005)* | JCF10830 | |
| Strain (*Schizosaccharomyces pombe*) | h⁻ leu1-32 ura4-D18 T2R1-4137::ura4 + bqt4::natMX6 | This study | JCF10833, 10834 | |
| Strain (*Schizosaccharomyces pombe*) | h⁻ swi6::LEU2 leu1-32 ura4-D18 T2R1-7921::ura4+ | *Kanoh et al. (2005)* | JCF10829 | |
| Strain (*Schizosaccharomyces pombe*) | h⁻ leu1-32 ura4-D18 T2R1-7921::ura4 + bqt4::natMX6 | This study | JCF11998-12002 | |
| Strain (*Schizosaccharomyces pombe*) | h⁻ leu1-32 ura4-D18 T2R1-7921::ura4+ | *Kanoh et al. (2005)* | JCF10832 | |
| Strain (*Schizosaccharomyces pombe*) | h⁺ lem2::kanMX6 | This study | JCF12178 | |

*Continued on next page*

Continued

| Reagent type (species) or resource | Designation | Source | Identifiers | Additional information |
|---|---|---|---|---|
| Strain (*Schizosaccharomyces pombe*) | $h^{90}$ epe1::natMX6 | This study | JCF10787 | |
| Strain (*Schizosaccharomyces pombe*) | epe1::natMX6 lem2::kanMX6 | This study | JCF12204 | |
| Strain (*Schizosaccharomyces pombe*) | $h^+$ lem2::kanMX6 dcr1::ura4 | This study | JCF12182 | |
| Strain (*Schizosaccharomyces pombe*) | $h^-$ epe1::naMX6t dcr1::ura4 | This study | JCF12209 | |
| Strain (*Schizosaccharomyces pombe*) | epe1::natMX6 dcr1::ura4 lem2::kanMX6 | This study | JCF12211 | |
| Strain (*Schizosaccharomyces pombe*) | $h^+$ cnp1-1 leu1 Rep41-ams2+[leu1+] | NBPR | JCF14580 | Yeast Genetic Resource Center (YGRC), Graduate School of Science, Osaka City University |
| Strain (*Schizosaccharomyces pombe*) | lem2::natMX6 cnp1-1 leu1 Rep41-ams2+[leu1+] | This study | JCF14588 | |
| Strain (*Schizosaccharomyces pombe*) | $h^+$ cnp1-1::ura4 + ura4-D18 | NBPR | JCF10266 | Yeast Genetic Resource Center (YGRC), Graduate School of Science, Osaka City University |
| Strain (*Schizosaccharomyces pombe*) | $h^-$ bqt4::nat leu1-32 ura4-D18 his3-D1 ade6-M210 | This study | JCF11933, 11934 | |
| Strain (*Schizosaccharomyces pombe*) | epe1::natMX6 dcr1::ura4 + bqt4::hygMX6 | This study | JCF12240 | |
| Strain (*Schizosaccharomyces pombe*) | $h^-$ bqt4::hygMX6 epe1::natMX6 | This study | JCF10811 | |
| Strain (*Schizosaccharomyces pombe*) | $h^+$ bqt4::hygMX6 dcr1::ura4+ | This study | JCF12180 | |
| Strain (*Schizosaccharomyces pombe*) | $h^-$ ade6-M210 leu1-32 ura4-D18 his3-D1 sid4-GFP-kanMX6 dh1L-tetO-ura+tetR-mCherry-nat | This study | JCF14534 | |
| Strain (*Schizosaccharomyces pombe*) | $h^-$ his7+::adh13pr-lacI-CFP C12::kanR-ura4+-lacOp sid4-mRFP-natR ade6-210 leu1-32 ura4-D18 lys1-131 | This study | JCF18924 | |
| Strain (*Schizosaccharomyces pombe*) | ade6-210 leu1-32 lys1-131 ura4-D18 sod2::kanr-ura4+-lacOp his7+::lacI-GFP cut11-RFP-hyg clr4::nat | This study | JCF11961 | |
| Strain (*Schizosaccharomyces pombe*) | lem2-GFP-Kan bqt4::nat pcp1-RFP-hph | This study | JCF12189 | |
| Strain (*Schizosaccharomyces pombe*) | lem2-GFP-Kan pcp1-RFP-hph | This study | JCF12190 | |
| Strain (*Schizosaccharomyces pombe*) | man1-GFP-kan ura4-D18 | This study | JCF12153 | |
| Strain (*Schizosaccharomyces pombe*) | $h^+$ man1 GFP-kan bqt4::nat ura4-D18 | This study | JCF12163 | |
| Strain (*Schizosaccharomyces pombe*) | $h^-$ ade6-M210 his3-D1 leu1-32 ura4-D18 nup107-GFP:ura4+ | This study | JCF11770 | |
| Strain (*Schizosaccharomyces pombe*) | nup107-GFP:ura4 + bqt4::nat | This study | JCF12177 | |
| Strain (*Schizosaccharomyces pombe*) | cut11-3xPK-GFP:ura4 + leu1-32 ura4-D18 | Lab stock (from Nurse lab) | JCF2924 | |
| Strain (*Schizosaccharomyces pombe*) | $h^+$ cut11 GFP-kan bqt4::nat ura4-D18 | This study | JCF12176 | |
| Strain (*Schizosaccharomyces pombe*) | $h^-$ bqt4::nat leu1-32 ura4-D18 his3-D1 ade6-M210 aur1R-GFP-bqt4ΔTM | This study | JCF12048, 12049 | |
| Strain (*Schizosaccharomyces pombe*) | lys1+-GFP-bqt4 bqt4::nat ura4-D18 ade6-M210 | This study | JCF11958 | |
| Strain (*Schizosaccharomyces pombe*) | imr1R(Nco1)::ura4 + swi7 GFP-kanR ura4-D18 leu1-32 ade6-M216 | This study | JCF18888 | |

*Continued on next page*

*Continued*

| Reagent type (species) or resource | Designation | Source | Identifiers | Additional information |
|---|---|---|---|---|
| Strain (*Schizosaccharomyces pombe*) | *imr1R(Nco1)::ura4 + swi7 GFP-kanR ura4-D18 leu1-32* | This study | JCF18890 | |
| Strain (*Schizosaccharomyces pombe*) | *imr1R(Nco1)::ura4 + ura4-D18 leu1-32 ade6-M216* | This study | JCF18892 | |
| Strain (*Schizosaccharomyces pombe*) | *imr1R(Nco1)::ura4 + ura4-D18 leu1-32* | This study | JCF18893 | |
| Strain (*Schizosaccharomyces pombe*) | *h⁺ his3D1leu1-32 ade6* | Lab stock | JCF901 | |
| Strain (*Schizosaccharomyces pombe*) | *h⁻ rad11-mCherry-kanR leu1-32 ura4-D18 ade6* | Lab stock | JCF1234 | |
| Strain (*Schizosaccharomyces pombe*) | *swi7-GFP::kanMX6 man1-GBP-mCherry-hph rad11-mCherry-Kan* | This study | JCF14598 | |
| Strain (*Schizosaccharomyces pombe*) | *taz1-mTurquoise2-natR man1-GBP-mCh-hphR ura4-D18 his3-D1 ade6* | This study | JCF18936 | |
| Strain (*Schizosaccharomyces pombe*) | *taz1-mTurquoise2-natR swi7-GFP-kanR man1-GBP-mCh-hphR ura4-D18 his3-D1 ade6* | This study | JCF18939 | |
| Strain (*Schizosaccharomyces pombe*) | *h⁹⁰ otr1R::ade6 + tel1L::his3 + tel2L::ura4 + ade6-210 his3-D1 leu1-32 ura4-D18* | Lab stock | JCF6712 | |
| Strain (*Schizosaccharomyces pombe*) | *h⁹⁰ otr1R::ade6 + tel1L::his3 + tel2L::ura4 + clr4::kanMX ade6-210 his3-D1 leu1-32 ura4-D18* | Lab stock | JCF6715 | |
| Strain (*Schizosaccharomyces pombe*) | *h⁹⁰ otr1R::ade6 + tel1L::his + 3 tel2L::ura + 4 bqt4::nat ade6-210 his3-D1 leu1-32 ura4-D18* | This study | JCF10763 | |
| Other | YES225 | Sunrise Science Products | Cat#2011–500 | |
| Other | PMG | Sunrise Science Products | Cat#2060–500 | |
| Other | EMM | Sunrise Science Products | Cat#2005–500 | |

## Yeast strains and media

Strains used in this study are described in the Key Resources Table. All media used were previously established (*Moreno et al., 1991*). Gene deletions and fluorophore-tag insertions were performed by the one-step gene replacement method, using appropriate plasmid templates (pFA6a-*kanMX6*, -*hygMX6*, and -*NatMX6* selection cassettes) as described previously (*Klutstein et al., 2015*). Functionality was tested for each tagged protein: Bqt4 function was checked by monitoring meiosis and telomere-NE localization, which were unaltered; however, GFP-Bqt4 containing cells show mild HU sensitivity (data not shown), suggesting that GFP-Bqt4 is a partial hypomorph. Lem2 functionality was tested by rescue of the thiabendazole sensitivity of *lem2Δ* cells (*Banday et al., 2016*), and Man1 by localization to the NE. Strain mating and random sporulation were used to combine multiple genetic mutations. Viability assays were performed by adding HU, MMS, and bleomycin at indicated concentrations to molten agar-containing rich yeast-extract media (YE5S; Sunrise Science Products, San Diego, CA) pre-cooled to 55°C. Overnight mid-log phase cell cultures (OD600 ~0.5) were serially diluted 5-fold, and stamped onto the plates containing media with the DNA damaging agents. Gene silencing assays were performed by stamping serially diluted cultures onto plates containing minimal medium (PMG, Sunrise Science Products) supplemented with appropriate amino acids and lacking the indicated nutrients.

## Live microscopy

Strain cultures were grown overnight at 32°C in 50 ml rich media to mid-log phase. After washing with EMM, cultures were diluted to original volume and the pellet from 1 ml of sample was resuspended in 20 µl of the imaging medium (1:2 YE5S-EMM). The resuspended cells (3 µl) were mounted on an agarose pad formed on a glass slide as described previously (*Hiraga et al., 2006*). Cells were imaged on a wide-field inverted DeltaVision (GE Healthcare) microscope with a 100X oil immersion objective (NA 1.42) and a Xenon lamp excitation source. Imaging was performed at 22–25°C. Images were captured and analyzed using SoftWoRx (GE Healthcare). Cells were imaged by capturing single time-point Z-stacks consisting of 25 images, with 0.25 µm between each Z-plane. The approach for analyzing positioning of the lacO/I dot within the nuclear zones was described previously (*Hediger et al., 2002*). To capture time-lapse images, cells were adhered to the bottom of a 35 mm glass bottom culture dishes (MatTek) as described previously (*Klutstein et al., 2015*), except that imaging was performed at 22–25°C and time-lapse intervals are indicated in the appropriate figure legends. The FRAP experiments were performed using a Zeiss LSM 710 Confocal system (Carl Zeiss Inc, Thornwood, NY) with a Zeiss Axiovert microscope, a 25 mW Argon 488 nm laser, and a 100x Plan-Neofluar 1.3 NA oil immersion objective. Digital images were 512 × 512 pixels and 12 bit. For FRAP, three laser lines of the Argon laser (458, 488, 514 nm) were used to bleach areas of the cell at 100% transmission and 10 iterations. The bleach was started after one acquisition scan and images were acquired at the indicated intervals.

## Super-resolution microscopy

SIM images were obtained with an Applied Precision OMX (GE Healthcare) using a 60 × 1.42 NA Olympus Plan Apo oil objective, and front illuminated sCMOS cameras (6.45 um pixel size). All SIM microscopy was performed at 22–23°C. Excitation sources consisted of a 488 nm laser (for GFP) or a 561 nm laser (for mCherry), and images were captured by alternating excitation using standard filters (FITC/AF488 and AF568/Texas Red). SIM reconstruction was done with SoftWorX, with a Wiener filter of 0.003. SIM images shown are single Z slices, digitally zoomed with bilinear interpolation using Fiji ImageJ (National Institutes of Health). To prepare cells for imaging, overnight cell cultures were grown (10 mL at OD595 ~ 0.5) and treated with 3.2% formaldehyde for 1 min before adding 0.1% Glutaraldehyde and incubating for 12 min at 25°C. The fixed cells were washed and treated with 0.6 mg/mL Zymolyase 100T at 36°C for 45 min, before treating with 1% Triton X-100 for 5 min at room temperature. Cells were washed and resuspended in 1 ml wash buffer (100 mM PIPES, 1 mM EGTA, 1 mM MgSO4 pH 6.9). To minimize photo-bleaching and amplify fluorescent signals, GFP and RFP boosters (ChromoTek, Germany) were added to 200 µl of the cells resuspended in wash buffer with added 1% BSA and 100 mM lysine hydrochloride. After incubation for 1 hr at room temperature, cells were washed and resuspended in 10 µl Vectashield (Vector Labs, Burlingame, CA). The cells (1 µl) were placed on a glass slide, covered with an acid washed coverslip, and sealed by nail polish before imaging.

## Chromatin immunoprecipitation

Cultures were grown overnight in 120 ml YE5S medium to mid-log phase and treated with formaldehyde (1% final conc.) for 10 min at room temperature with occasional swirling. As described in *Figure 6C*, 37.5 ml of the spike culture was mixed with 112.5 ml of each sample before treating with formaldehyde. Treated cells were washed with 50 mL refrigerated Milli-Q (Millipore) water. Cells were centrifuged at 2000 rpm for 2 min, resuspended in 1 ml water and centrifuged before adding 1 ml Zirconium beads to the pellet. The pellet and beads mixture was flash frozen in liquid nitrogen and stored at −80°C in 2 ml screw-cap shatter proof tubes. The frozen pellet was resuspended in 500 µl lysis buffer: 50 mM Hepes pH 7.4, 140 mM NaCl, 1 mM EDTA, 1% Triton X-100, 0.1% Sodium Deoxycholate, 1% PMSF and EDTA-free protease inhibitor cocktail (Roch) and lysed using FastPrep (MP Biomedical; setting: 5 × 30 s cycles at 4.5 m/s). Each lysate was recovered in a 5 ml snap-cap tube by centrifugation at 1500 rpm for 3 min and sonicated using the Covaris E210 instrument. Sonication settings used were: 20%, 8 intensity, 200 cycles per burst for 12 min. The sonicated lysates were clarified by centrifugation at 14000 rpm for 10 min. For input DNA, 20 µl of the clarified lysate was set aside. The lysates were incubated with 5 µg H3K9me2 (abcam #1220) antibody overnight at 4°C. Magnetic protein-G beads (Invitrogen, Waltham, MA) were added to the lysate/antibody

mixture and incubated for 4 hr before washing and eluting the immunoprecipitate from the beads. Elution by boiling and reverse cross-linking was performed using the ChIP Elute Kit (Clontech) to obtain ssDNA templates for Next-Gen library preparation.

## Next-Generation sequencing

The ssDNA samples prepared above were quantified using the Quibit ssDNA Assay Kit (molecular probes, life technologies), and 5–10 ng of each sample was used for library preparation. Libraries were prepared using DNA SMART ChIP-Seq Kit (Clontech) and Illumina indexed oligomers. Following PCR amplifications (15 cycles), libraries were size selected and purified using Agencourt AMPure XP beads (200–400 bp range). Each library was validated using Quibit 2.0 Fluorometer for concentration, Agilent 2100 Bioanalyzer for fragment size, and NEBNext Library Quant Kit (New England BioLabs) for adaptor presence. Typical library yield was 10–40 ng with average size of 400 bp. Finally, libraries from 12 samples were pooled in equimolar ratio (10 nM) and submitted for sequencing using Illumina HiSeq2500 Rapid Run mode (GENEWIZ, South Plainfield, NJ). A total of ~140 million 50 bp reads were obtained per pooled sample. The reads for each sample in the pool were separated per the corresponding Illumina indexes (% of >= Q30 bases: 97.85; mean quality score: 38.63), before subjecting to bioinformatics analysis.

## Bioinformatic analysis

As required by the library construction procedure, three bases at the 5' ends of the reads were trimmed. The 47 bp post-trimmed reads were aligned to the reference genome using bowtie2, allowing for 50 alignments per repetitive region. The original reference sequence for Chr II contains a partially deleted sequence of the *mat* locus; therefore, we customized the reference genome by replacing the deleted *mat* region with a contig of the $h^{90}$ configuration of the *mat2P-mat3M* region (as described on 'http://www.pombase.org/status/mating-type-region'; accessed July.2017). Before normalization, the resulting aligned SAM files were converted to sorted and indexed BAM files. Normalization was performed using a Python script that sequentially executed: 1- Get the number of total aligned reads, reads covering *ade6* locus (chr III: 1316291–1318035), and reads covering a euchromatin region adjacent to *ade6* locus (chr III: 1304000,1380000); 2- Calculate a normalization scale factor: 10 / (median of reads covering each base within the *ade6* locus – median of reads covering each base in the neighboring euchromatic region); 3- Apply the scale factor to the aligned genomic reads (BAM files) using the genomecov function of the BEDTools suit to produce a bedgraph file. To generate the ribbon-line coverage plots shown in the above figures, an R script was created that imports the bedgraph files and, using a 250 bp sliding window, calculates the median, maximum and minimum of reads from strains with identical genotypes. The scripts used for this study have been deposited at 'https://www.elucidaid.com/repository/ebrahimih17'.

## Acknowledgements

We thank Michael Lichten, Shiv Grewal and our laboratory members for discussions; Robin Allshire, Yasushi Hiraoka and Masamitsu Sato for generous gifts of strains and reagents; Yoshiyuki Wakabayashi for help with design of the ChIP-Seq protocol; Susan Garfield for help with the confocal microscopy/FRAP experiments; and Patrina Pellett for help with OMX imaging setup.

## Additional information

### Funding

| Funder | Grant reference number | Author |
| --- | --- | --- |
| National Cancer Institute | Intramural funding | Hani Ebrahimi<br>Hirohisa Masuda<br>Julia Promisel Cooper |

The funders had no role in study design, data collection and interpretation, or the decision to submit the work for publication.

## Author contributions
Hani Ebrahimi, Conceptualization, Methodology, Investigation, Writing—original draft, Software, Data curation; Hirohisa Masuda, Conceptualization, Investigation, Methodology, Writing—reviewing and editing, Data curation; Devanshi Jain, Conceptualization, Methodology, Investigation, Performed the initial screen that identified Bqt4 as required for HAATI maintenance, and analyzed viability and damage sensitivity of cells lacking Bqt; Julia Promisel Cooper, Conceptualization, Investigation, Methodology, Writing- original draft, Writing—review and editing, Funding acquisition, Formal analysis

## Author ORCIDs
Julia Promisel Cooper http://orcid.org/0000-0003-2171-2587

## Decision letter and Author response
Decision letter https://doi.org/10.7554/eLife.32911.023
Author response https://doi.org/10.7554/eLife.32911.024

## Additional files

### Supplementary files
• Transparent reporting form
DOI: https://doi.org/10.7554/eLife.32911.021

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
