## [Decision Letter]

Thank you for submitting your article "Distinct 'safe zones' at the nuclear envelope ensure robust replication of heterochromatic chromosome regions" for consideration by *eLife*. Your article has been favorably evaluated by Jessica Tyler (Senior Editor) and three reviewers, one of whom is a member of our Board of Reviewing Editors. The following individuals involved in review of your submission have agreed to reveal their identity: Marc R Gartenberg (Reviewer #2); Mikel Zaratiegui (Reviewer #3).

After a productive online discussion, a very positive and coherent view of the paper has emerged. The text below should guide you to prepare a revised submission.

All three reviewers were impressed by the experiments documenting non-overlapping domains near the nuclear envelope. Also, the functional importance of these separate domains for efficient replication of heterochromatic areas was, by and large, considered an important advance in our knowledge in this area. The new pol α tethering system in particular was mentioned to be creative and inspiring, allowing for completely new angles to be explored. These major points, taken together, make for a very strong and important contribution to the field. In fact, many researchers in annexed fields will be interested in these studies as well and therefore a very high level of interest is expected. For these reasons, the reviewers felt that your paper may be of the level expected at *eLife* and, with the few revisions added as noted below, could become acceptable.

Essential points requiring new experiments for re-submission:

1) Please verify for co-imaging localization of Lem2 vs. Man1 areas with this setup. This would add important weight to the core argument of specific areas under the NE.

2) The authors show that the chimeric Man1-GBP-mCherry causes pol α to localize to the nuclear periphery. They do not show that any specific telomere or centromere moves to a Man1 micro-domain and out of its respective Lem2 and Bqt4 micro-domain. This seems doable. It would show that they actually have forced relocalization of specific heterochromatin domains, and not just bulk pol α.

3) Figure 6F. The correct comparison is Pol α-GFP/Man1+ Vs Pol α-GFP/Man1GBP. From what the text says, it seems like the sample labeled as WT is pol α+/Man1-GBP. But the tag more likely to affect heterochromatin by itself is Pol α-GFP; Pol α was originally identified in *pombe* as *swi7* because its mutation impairs mating type switching (Singh and Klar 1993 PMID 8423854) and was later recognized as a suppressor of variegation (Nakayama and Grewal 2001 PMID 11387218). I could not find a PEV test in the paper where this *swi7*-GFP allele is described (Natsume at al 2008 PMID 18493607). A simple RTPCR for centromeric transcripts showing that they are not upregulated in the *swi7*-GFP strain would be sufficient to exclude the possibility of an effect of the tag.

Other possibilities, more labor intensive but also more convincing, would be a PEV assay (with a *ura4*+ reporter in *otr* or *imr*) or an H3K9me2 ChIP. However, if these alternatives take more than two months to complete, they would not be essential for the paper.

4) The correct control was not done for Figure 5—figure supplement 1A. They omitted wt GFP-Bqt4.

Next here I list a number of issues and requests that the reviewers mentioned and that can easily be addressed by editorial changes. In other words, we do not expect more experimental evidence for these points, but the manuscript text hopefully will be adjusted to reflect the raised issues:

1) All three reviewers felt that the Discussion loses its punch by going on so long. In their opinion, the whole manuscript and the Discussion is overly focused on the notion of replication/transcription conflicts, which were not extensively examined experimentally. Given that obtaining direct experimental evidence for such conflicts may be unreasonably labour- and time-intensive, they suggest toning that discussion down accordingly and shortening the Discussion overall.

2) Some places would benefit from a better quantitative description:

- The Bqt4 dependent Lem2 spreading in the FRET assays

- In the first paragraph of the subsection “Bqt4 tethers telomeres and the *mat* locus specifically while they are replicating”, it is said that Taz1 – Bqt4 interaction is better than Taz1 – Lem2. The Bqt4-Taz1 colocalization should be quantified, like that presented of Mis6 and Lem2 in Figure 4—figure supplement 1B.

3) Why the different behavior/localization of Telo1L vs. Taz1 (which also paints telomeres). Compare: Figure 4C early/mid S: significant drop of NE localization WT vs. bqt4- to: Figure 4D early/mid S: No difference in NE localization WT vs. bqt4-.

4) Figure 5—figure supplement 1A: although mentioned in the text, there is no MMS panel presented.

5) Figure 7G has no scale and therefore is difficult to compare to Figure 6F.

6) There is no Figure 9 (Model), but it is discussed in the Discussion. The reviewers actually would like to see that figure included.

7) Centromeres colocalize with Lem2 spot under the SPB (Figure 4A). Pericentric heterochromatin (spanning 100kb around Cen3) often localizes beyond the massive Lem2 spot (Figure 4—figure supplement 1C). I think the authors are implying pericentric heterochromatin interacts with micro-domains of Lem2 that are distinct from the massive Lem2 spot. This seems at odds with the sentence: "The intense focus of Lem2 detectable in this region (Figure 4A) is consistent with ChIP experiments showing an interaction between Lem2 and pericentric heterochromatin (Barrales et al., 2016; Tange et al., 2016)." Moreover, I think the Barrales and Tange papers showed Lem2 association with CenpA chromatin but not broader pericentric heterochromatin.

8) "Our microscopy shows distinct localization patterns for centromere cores and pericentric regions, with the centromeric cores colocalizing with Lem2 beneath the SPB and the pericentromeres colocalizing with non-SPB-adjacent Lem2." To my knowledge, the authors never showed this. They showed in Figure 4—figure supplement 1C that the pericentromeres were not at the SPB. They did not show that this chromatin abutted non-SPB-adjacent Lem2.

9) The description of the cell cycle in the fourth paragraph of the subsection “Bqt4 tethers telomeres and the *mat* locus specifically while they are replicating” will confuse readers not familiar with *pombe*. Make this description accessible to a broad readership.

10) This is only a minor comment but it is actually quite important. Throughout, the authors refer to H3K9me. However, their experiments are done with an antibody against H3K9me2. They should make this clear, because while H3K9me2 remains a good indicator of heterochromatin formation and RNAi activity, there are clear functional differences between H3K9me2 and H3K9me3 that were recently described (Jih et al. 2017, Moazed lab PMID 28682306).

11) Throughout, the microscopy images have no sizing bars.

12) Throughout, most comparisons are not statistically analyzed. This may affect interpretation of some conclusions, like in Figure 4C and D. Is Telomere peripheric localization in early/midS vs. lateS/G2 in bqt4D significant? Bqt4D vs. WT in early/midS significant?

13) Figure 5—figure supplement 1A. Is the GFP-bqt4+ strain not sensitive to HU? This would rule out an effect of the GFP tag on Bqt4.

14) Subsection “Bqt4 and Lem2 microenvironments regulate telomeric and centromeric heterochromatin”, end of fourth paragraph: The pericentric H3K9me data is presented in Figure 7—figure supplement 1C, not 1A.

15) Figure 8B, in subsection “Lem2 and Bqt4 become crucial for viability in the absence of Dcr1”, first paragraph: *lem2*+ deletion greatly reduces the viability of *cen1-1* ts. I can't find the *lem1D/cen1-1* data in this figure or in the supplemental, only the *lem1D/cen1-1*/pnmt41-AMS and *cen1-1*/pnmt41-AMS.

16) "[…] cause extra trouble in HU or MMS". As much as I appreciate the colloquial phrasing, this is too vague. Please be more specific about how the authors propose that MMS or HU in combination with *bqt4Δ* could lead to an untenable situation. Are the DNA repair pathways overwhelmed by exacerbated endogenous replication stress (in *bqt4Δ*) along with that caused by exogenous agents (HU/MMS)?

17) Introduction, third paragraph. Here please cite Castel et al., 2014, which is the one showing evidence of Dcr1 releasing RNAPol II from tDNA and rDNA.

---

## [Author Response]

Essential points requiring new experiments for re-submission:1) Please verify for co-imaging localization of Lem2 vs. Man1 areas with this setup. This would add important weight to the core argument of specific areas under the NE.We agree that simultaneous Lem2/Man1 localization is important for completing the picture and have added this data to the new Figure 1. This new data had to be captured via conventional DeltaVision imaging rather than SIM, since the amplified td-tomato tag, the brightest we could obtain for Man1, gave insufficient SIM signal (probably due to the fixation needed for SIM). Nonetheless, the Man1-tdTomato/Lem2-GFP combination is easily visible by live analysis, which shows a clear separation of Lem2 and Man1 domains. We have now quantified the colocalization of all pairwise combinations of visualized proteins (Figure 1—figure supplement 1).2) The authors show that the chimeric Man1-GBP-mCherry causes pol α to localize to the nuclear periphery. They do not show that any specific telomere or centromere moves to a Man1 micro-domain and out of its respective Lem2 and Bqt4 micro-domain. This seems doable. It would show that they actually have forced relocalization of specific heterochromatin domains, and not just bulk pol α.We agree, and had tried with limited success to visualize telomeres and centromeres in Man1-GBP-Cherry settings with and without Polα-GFP. The tagging combinations made this experiment difficult. Now, by tagging Taz1 with mTurquoise and using the mCherry tag on Man1-GBP, we have been able to do this experiment – both representative images and a graph of the results are now shown in Figure 6—figure supplement 2C-E. The data show clearly that the Man1-GBP/Polα-GFP combination moves telomeres from Man1-poor regions to Man1-rich regions. The effect is modest as expected (since this relocalization only occurs transiently, while a given telomere replicates) but significant.3) Figure 6F. The correct comparison is Pol α-GFP/Man1+ vs. Pol α-GFP/Man1GBP. From what the text says, it seems like the sample labeled as WT is pol α+/Man1-GBP. But the tag more likely to affect heterochromatin by itself is Pol α-GFP; Pol α was originally identified in pombe as swi7 because its mutation impairs mating type switching (Singh and Klar 1993 PMID 8423854) and was later recognized as a suppressor of variegation (Nakayama and Grewal 2001 PMID 11387218). I could not find a PEV test in the paper where this swi7-GFP allele is described (Natsume at al 2008 PMID 18493607). A simple RTPCR for centromeric transcripts showing that they are not upregulated in the swi7-GFP strain would be sufficient to exclude the possibility of an effect of the tag.Other possibilities, more labor intensive but also more convincing, would be a PEV assay (with a ura4+ reporter in otr or imr) or an H3K9me2 ChIP. However, if these alternatives take more than two months to complete, they would not be essential for the paper.

Good point – we have addressed this by assessing the effect of the Polα-GFP tag on silencing in the *imr* region of the centromere (where an inserted *ura4+* marker is silenced). The results are very clear in showing that the tag has no impact on silencing at *imr*, which is completely intact, as shown in the new Figure 6—figure supplement 1.

4) The correct control was not done for Figure 5—figure supplement 1A. They omitted wt GFP-Bqt4.

Thank you for catching this. We have repeated the experiment with wt GFP-Bqt4 and present the new data in Figure 5—figure supplement 1. The GFP-Bqt4 rescues the MMS sensitivity of *bqt4Δ* cells but not the HU sensitivity, suggesting that GFP-Bqt4 is a partial hypomorph. This was a surprise as the Hiraoka lab had found that while C-terminal tags on Bqt4 abolish function, the N-terminally tagged Bqt4 we are using conferred no phenotype. We have now altered the Materials and methods section to reflect this HU sensitivity, and removed the ΔTM data for HU.

Next here I list a number of issues and requests that the reviewers mentioned and that can easily be addressed by editorial changes. In other words, we do not expect more experimental evidence for these points, but the manuscript text hopefully will be adjusted to reflect the raised issues:1) All three reviewers felt that the Discussion loses its punch by going on so long. In their opinion, the whole manuscript and the Discussion is overly focused on the notion of replication/transcription conflicts, which were not extensively examined experimentally. Given that obtaining direct experimental evidence for such conflicts may be unreasonably labour- and time-intensive, they suggest toning that discussion down accordingly and shortening the Discussion overall.We agree. We have shortened the Discussion significantly and tried to streamline and clarify.2) Some places would benefit from a better quantitative description:- The Bqt4 dependent Lem2 spreading in the FRET assays

The Lem2 FRAP data was quantified in Figure 3C, but note that FRAP cannot be performed on signals as weak as the peripheral (i.e., not SPB-adjacent) Lem2 signals. Therefore, we are not confident in quantifying spreading around the periphery and prefer to stick with a qualitative description of this (hopefully I have not misunderstood your point).

- In the first paragraph of the subsection “Bqt4 tethers telomeres and the mat locus specifically while they are replicating”, it is said that Taz1 – Bqt4 interaction is better than Taz1 – Lem2. The Bqt4-Taz1 colocalization should be quantified, like that presented of Mis6 and Lem2 in Figure 4—figure supplement 1B.

We have quantified these colocalizations in the new Figure 3G.

3) Why the different behavior/localization of Telo1L vs. Taz1 (which also paints telomeres). Compare: Figure 4C early/mid S: significant drop of NE localization WT vs. bqt4- to: Figure 4D early/mid S: No difference in NE localization WT vs. bqt4-.Thank you for asking us to clarify this. The Tel1L marker (a lacO/I array at the *sod2+* locus) is over 50kb in from the telomere and may often replicate earlier than the actual telomeres (to which Taz1 binds). We have clarified this in the text.4) Figure 5—figure supplement 1A: although mentioned in the text, there is no MMS panel presented.We apologize for this omission – we have fixed it (please see point 4 above as well).5) Figure 7G has no scale and therefore is difficult to compare to Figure 6F.We have added the scale values.6) There is no Figure 9 (Model), but it is discussed in the Discussion. The reviewers actually would like to see that figure included.

We apologize for this omission We have added the model figure that we prepared for the original manuscript (Figure 9).

7) Centromeres colocalize with Lem2 spot under the SPB (Figure 4A). Pericentric heterochromatin (spanning 100kb around Cen3) often localizes beyond the massive Lem2 spot (Figure 4—figure supplement 1C). I think the authors are implying pericentric heterochromatin interacts with micro-domains of Lem2 that are distinct from the massive Lem2 spot. This seems at odds with the sentence: "The intense focus of Lem2 detectable in this region (Figure 4A) is consistent with ChIP experiments showing an interaction between Lem2 and pericentric heterochromatin (Barrales et al., 2016; Tange et al., 2016)." Moreover, I think the Barrales and Tange papers showed Lem2 association with CenpA chromatin but not broader pericentric heterochromatin.In the Barrales et al. paper, Lem2 was identified as a silencing factor for the *imr* region, but also shown to regulate silencing at the *cen-dg* repeats which are more distal; these effects of Lem2 on silencing were small. The Tange paper also shows reduced H3K9me2 at both *dg/dh* repeats and *imr::ura4* in the absence of Lem2. At the same time, both papers show Lem2 binding specifically to the central core. Our work is consistent with these observations in showing a role for Lem2 in CenpA loading at the central core (independent of the Bqt4-dependent mobilization of Lem2 away from the SPB) as well as a modest effect on pericentric silencing, likely mediated by transient (S-phase) Lem2-pericentromere associations that were not picked up by the ChIP experiments of the previous papers. We have corrected our sentence to say “consistent with ChIP experiments showing an interaction between Lem2 and centromeres”8) "Our microscopy shows distinct localization patterns for centromere cores and pericentric regions, with the centromeric cores colocalizing with Lem2 beneath the SPB and the pericentromeres colocalizing with non-SPB-adjacent Lem2." To my knowledge, the authors never showed this. They showed in Figure 4—figure supplement 1C that the pericentromeres were not at the SPB. They did not show that this chromatin abutted non-SPB-adjacent Lem2.We have altered the text to clarify that the idea of colocalization between pericentromeres and non-SPB-localized Lem2 is an inference rather than a result.9) The description of the cell cycle in the fourth paragraph of the subsection “Bqt4 tethers telomeres and the mat locus specifically while they are replicating” will confuse readers not familiar with pombe. Make this description accessible to a broad readership.We have simplified this description.10) This is only a minor comment but it is actually quite important. Throughout, the authors refer to H3K9me. However, their experiments are done with an antibody against H3K9me2. They should make this clear, because while H3K9me2 remains a good indicator of heterochromatin formation and RNAi activity, there are clear functional differences between H3K9me2 and H3K9me3 that were recently described (Jih et al. 2017, Moazed lab PMID 28682306).We have altered the text to use ‘H3K9me’ for instances in which we refer to me2 and me3 collectively, while we specify H3K9me2 for experiments using the corresponding antibody. In our original Introduction, we specified that H3K9me refers to both me2 and me3.11) Throughout, the microscopy images have no sizing bars.We have added scale bars throughout.12) Throughout, most comparisons are not statistically analyzed. This may affect interpretation of some conclusions, like in Figure 4C and D. Is Telomere peripheric localization in early/midS vs. lateS/G2 in bqt4D significant? Bqt4D vs. WT in early/midS significant?Thank you for flagging this up – we have added the relevant p values to Figure 4 and a statistics table (Figure 4—table supplement 1).13) Figure 5—figure supplement 1A. Is the GFP-bqt4+ strain not sensitive to HU? This would rule out an effect of the GFP tag on Bqt4.

Please see point 4 above. Our new experiments show that *GFP-bqt4+* is insensitive to MMS but mildly sensitive to HU.

14) Subsection “Bqt4 and Lem2 microenvironments regulate telomeric and centromeric heterochromatin”, end of fourth paragraph: The pericentric H3K9me data is presented in Figure 7—figure supplement 1C, not 1A.Thank you – fixed.15) Figure 8B, in subsection “Lem2 and Bqt4 become crucial for viability in the absence of Dcr1”, first paragraph: lem2+ deletion greatly reduces the viability of cen1-1 ts. I can't find the lem1D/cen1-1 data in this figure or in the supplemental, only the lem1D/cen1-1/pnmt41-AMS and cen1-1/pnmt41-AMS.Thank you for prompting us to clarify this. We cannot obtain double mutant *lem2 cnp1-1* cells; only upon Ams2 overexpression can this mutant combination be grown at all. We have altered the text to explain this.16) "[...] cause extra trouble in HU or MMS". As much as I appreciate the colloquial phrasing, this is too vague. Please be more specific about how the authors propose that MMS or HU in combination with bqt4Δ could lead to an untenable situation. Are the DNA repair pathways overwhelmed by exacerbated endogenous replication stress (in bqt4Δ) along with that caused by exogenous agents (HU/MMS)?We have altered this text, to suggest that repetitive regions with a tendency to stall replication (likely due to spurious secondary structure formation on unwinding) are more vulnerable to the replication stress conferred by HU and MMS. This could be due to the enhanced lifetime of the unwound region, to overwhelming of the DNA repair machinery as suggested by the reviewer, or to the collapse of forks already stalled by repetitive sequences.17) Introduction, third paragraph. Here please cite Castel et al., 2014, which is the one showing evidence of Dcr1 releasing RNAPol II from tDNA and rDNA.

Done. We had intended to include Castel et al. from the outset – thank you for catching that.